# Elevated moisture stimulates carbon loss from mineral soils by releasing protected organic matter

Wenjuan Huang[1] & Steven J. Hall [1]

Moisture response functions for soil microbial carbon (C) mineralization remain a critical uncertainty for predicting ecosystem-climate feedbacks. Theory and models posit that C mineralization declines under elevated moisture and associated anaerobic conditions, leading to soil C accumulation. Yet, iron (Fe) reduction potentially releases protected C, providing an under-appreciated mechanism for C destabilization under elevated moisture. Here we incubate Mollisols from ecosystems under $C_3/C_4$ plant rotations at moisture levels at and above field capacity over 5 months. Increased moisture and anaerobiosis initially suppress soil C mineralization, consistent with theory. However, after 25 days, elevated moisture stimulates cumulative gaseous C-loss as $CO_2$ and $CH_4$ to >150% of the control. Stable C isotopes show that mineralization of older $C_3$-derived C released following Fe reduction dominates C losses. Counter to theory, elevated moisture may significantly accelerate C losses from mineral soils over weeks to months—a critical mechanistic deficiency of current Earth system models.

[1] Department of Ecology, Evolution, and Organismal Biology, Iowa State University, 251 Bessey Hall, Ames, IA 50011, USA. Correspondence and requests for materials should be addressed to S.J.H. (email: stevenjh@iastate.edu)

Microbial decomposition of organic matter contributes a large efflux of carbon (C) from soils that critically impacts the global C cycle[1]. Heterotrophic microbial respiration is fundamentally influenced by soil moisture[2, 3]. Climate change has altered the spatiotemporal distribution of precipitation, often resulting in dry ecosystems getting drier and wet regions getting wetter, with a greater proportion of precipitation in extreme precipitation events and pronounced seasonal shifts in water balance[4]. The response of soil heterotrophic microbial respiration to changes in moisture under climate change may significantly impact the C balance of terrestrial ecosystems, but hinges on biogeochemical mechanisms controlling C availability to microbes.

Maximum organic matter decomposition is thought to occur at an intermediate moisture content that optimizes both oxygen ($O_2$) supply from the atmosphere and C substrate diffusion through soil water[2, 3]. Accordingly, short-term heterotrophic carbon dioxide ($CO_2$) production typically peaks under moist, but non-saturated conditions[5]. A substantial body of work has addressed the relationship between sub-optimal moisture and drought stress on heterotrophic $CO_2$ production[6]. However, the respiratory response at the other end of the moisture curve, where high moisture limits $O_2$ availability, has received less attention[7]. Elucidating the mechanisms controlling the response of heterotrophic microbial activity to increased moisture remains a critical gap for predicting C cycle feedbacks to climate change.

Reactive soil minerals, and iron (Fe) phases in particular, play a critical role in protecting soil C from microbial decomposition[8]. For example, hydrophilic and carboxylic C that is readily assimilated by microbes can be stabilized by Fe oxides via sorption and co-precipitation[9]. The dominant pools of mineral-associated organic C in many surface soils turn over on decadal timescales, despite the persistence of smaller pools of that cycle over centennial to millennial timescales[10–12]. However, the biogeochemical processes that drive the release and subsequent decomposition of mineral-associated organic C have received less attention (for example, see Keiluweit et al.[13]) than mineral protection of C[8, 14]. In particular, protective associations between Fe mineral phases and soil organic C may be vulnerable to moisture-sensitive redox dynamics[14–16]. Significant portions of C protected by Fe complexation under aerobic conditions (accounting for up to 40% of total soil organic C[9, 17]) could potentially be released and decomposed following Fe reduction.

We propose that the short-term (i.e., hours–days) suppression of soil respiration under elevated moisture described in previous studies[2, 3, 5] may be counteracted by the release and mineralization of organic C that is bound in association with Fe oxide minerals over longer timescales (weeks–months). Anaerobic (reducing) conditions may increase C availability in soluble and colloidal forms following bacterial dissimilatory Fe reduction[15, 16, 18]. This can occur directly due to C release from C-Fe associations, as well as from pH-mediated increases in C solubility driven by Fe reduction[15, 19–21]. Iron reduction can potentially occur rapidly (hours–days) following elevated moisture and/or labile C inputs in many terrestrial soils[11, 22, 23], and most terrestrial soils contain significant stocks of reducible Fe oxides[24]. However, to our knowledge, the importance of Fe-mediated release of colloidal or dissolved organic C (DOC) in mediating the overall response of heterotrophic respiration to elevated soil moisture has not been examined.

It remains unclear whether potential increases in DOC availability following Fe reduction are sufficient to offset kinetic and thermodynamic constraints on decomposition that accompany $O_2$ limitation. Anaerobiosis is thought to decrease rates of decomposition relative to aerobic conditions[25]. Activities of soil hydrolytic enzymes that proximately control soil organic matter decomposition are thought to decrease under anaerobic conditions because of decreased enzyme production and inhibition from phenolic substances[26, 27]. However, temporary anaerobiosis per se does not necessarily inhibit heterotrophic activity when hydrolysable C or monomers are available[28]. Release of biochemically labile C following Fe reduction could potentially offset upstream limitations on macromolecular decomposition due to anaerobiosis.

Here we test the effects of high soil moisture and associated reducing conditions on soil C mineralization in C-rich former grassland and wetland soils spanning three topographic positions (ridge, footslope, and depression) in Iowa, USA. Large seasonal fluctuations in moisture and water level commonly occur in these soils, even with artificial drainage systems[29, 30]. In the most poorly drained soils, moisture often increases in spring, remains high for several months, and then decreases in mid-summer due to increased evapotranspiration[31]. Although soils on footslopes and depressions are more prone to periodic flooding, soils on ridges also experience seasonal fluctuations in surface moisture and water table depth[32]. Thus, pronounced seasonal variability in soil moisture could potentially drive Fe reduction and oxidation and associated C dynamics across the landscape. These ecosystems are presently cultivated with corn (a $C_4$ plant) and soybean (a $C_3$ plant) rotated annually, representative of the dominant land cover across the North American Corn Belt. The natural stable C isotope ($\delta^{13}C$) labels provided by this rotation allow us to quantify solubilization and mineralization of newer (most recent growing season) vs. older C sources as affected by elevated moisture and Fe redox cycling.

In this study, we assess the effects of soil moisture on soil $CO_2$ and $CH_4$ production and their $\delta^{13}C$ values at three soil moisture levels: field capacity, intermediate, and saturation (51%, 77%, and 99% water-filled pore space (WFPS), respectively). Saturated soils are allowed to drain slowly after 82 days, analogous to the hydroperiod of seasonal wetlands in our region, while the field capacity and intermediate treatments remain static. Shorter-term moisture fluctuations (days) are also of interest in these ecosystems, but here we seek to assess biogeochemical impacts of elevated moisture over weeks–months as an end member to challenge conceptual models of heterotrophic activity developed over shorter timescales. Carbon mineralization is initially suppressed under saturated conditions, consistent with long-standing theory. However, following a lag period, the release of Fe mineral-associated organic C as DOC stimulates subsequent C mineralization as $CO_2$ and $CH_4$ (Supplementary Fig. 1). The loss of older $C_3$-derived C increases under reducing conditions relative to field capacity. We suggest that reducing conditions can accelerate C loss in mineral soils by facilitating microbial access to previously protected C sources.

## Results

**Carbon mineralization over time.** We found that impacts of soil moisture on C mineralization ($CO_2$ and $CH_4$ production) varied significantly over time, with initial suppression followed by subsequent enhancement in both the intermediate and saturated/drained treatments relative to the field capacity treatment, hereafter termed the control (Fig. 1 and Supplementary Figs. 2 and 3). Impacts of moisture treatments were generally similar among soils from different topographic positions when C mineralization was expressed on a soil mass basis, an appropriate metric for modeling studies (for comparison, results normalized by soil C are shown in Supplementary Fig. 3). During the first 10 days of the experiment, $CO_2$ production was consistently depressed in the saturated and the intermediate treatments relative to the control in all three soils ($p < 0.01$ for all three soils) (Supplementary

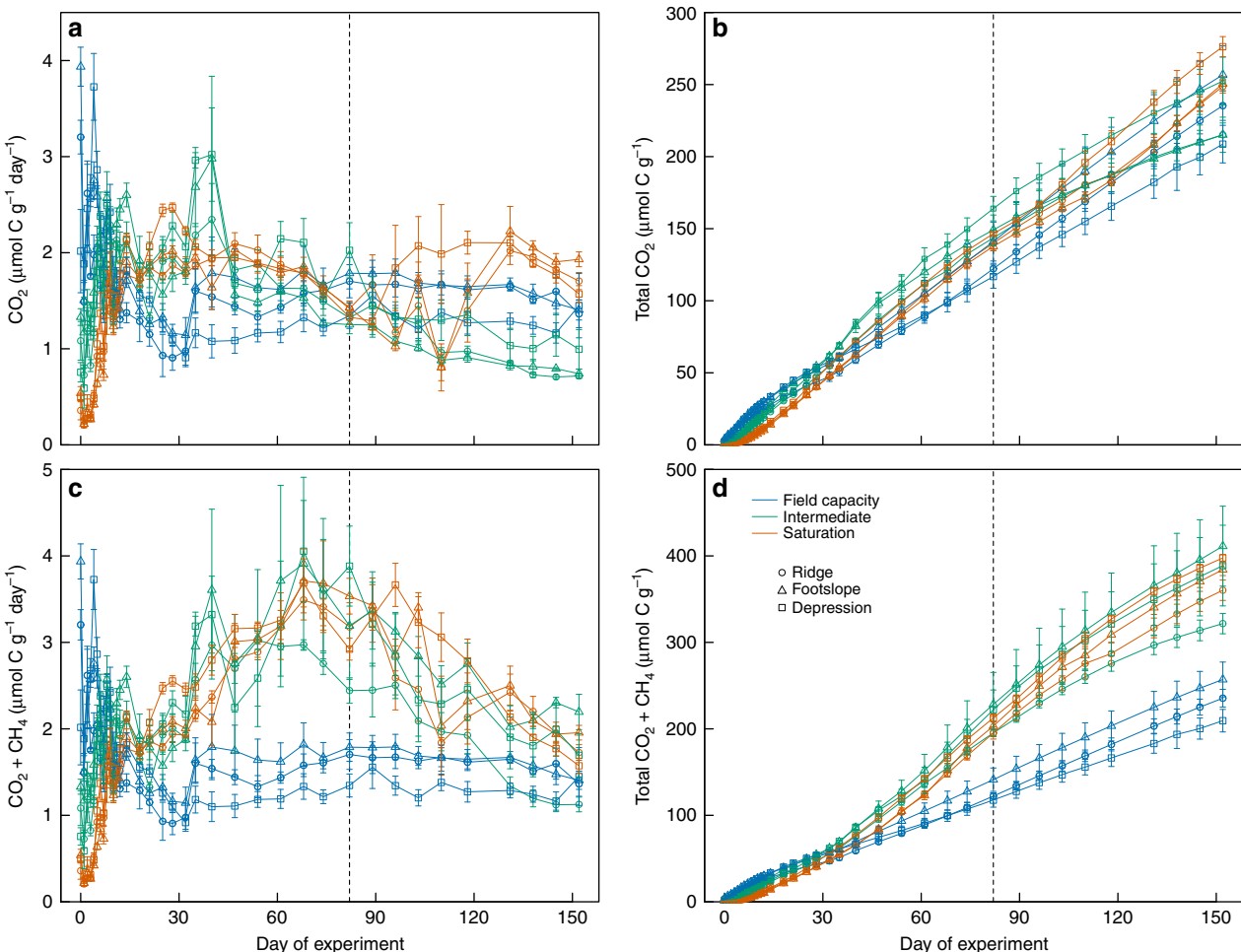

**Fig. 1** Carbon mineralization from three Mollisols incubated under moisture levels at and above field capacity. **a** $CO_2$ production rate; **b** cumulative $CO_2$ production; **c** total C mineralization rate ($CO_2 + CH_4$); **d** cumulative total C mineralization ($CO_2 + CH_4$). The vertical dashed line indicates when gradual drainage was initiated in the saturated soils. The error bars indicate s.e.m. ($n = 4$)

Fig. 2a). At 25 days, cumulative $CO_2$ production from all three soils remained significantly lower in the saturated treatment than in the intermediate treatment ($p < 0.05$ for the ridge and depression soils and $p < 0.01$ for the footslope soil) and the control ($p < 0.05$ for the ridge soil and $p < 0.01$ for the footslope and depression soils) (Supplementary Fig. 2b). However, the relationship between moisture and $CO_2$ production was reversed between 25 and 82 days. During this period, $CO_2$ production from the ridge and depression soils in both the intermediate and saturated treatments became significantly higher than the control ($p < 0.01$), and $CO_2$ production from the footslope soil became statistically equivalent among the three moisture treatments (Fig. 1a). Cumulative $CO_2$ production at 82 days was similar among the three moisture treatments in the ridge ($136 \pm 4 \, \mu mol$ $C \, g^{-1}$ soil) and footslope soils ($143 \pm 5 \, \mu mol \, C \, g^{-1}$ soil), and was significantly higher in the intermediate ($164 \pm 8 \, \mu mol \, C \, g^{-1}$ soil; $p < 0.01$) and saturated ($147 \pm 1 \, \mu mol \, C \, g^{-1}$ soil; $p < 0.05$) treatments than in the control ($117 \pm 8 \, \mu mol \, C \, g^{-1}$ soil) in the depression soil (Fig. 1b).

The natural hydroperiod of hydric soils in this study area (the southern Prairie Pothole region) is often characterized by spring saturation followed by gradual drainage due to seasonal trends in precipitation and evapotranspiration[31]. Therefore, after 82 days, the saturated treatment was allowed to slowly drain over the subsequent 70 days to simulate seasonal variation in moisture under field conditions. Moisture in the field capacity and

intermediate treatments remained consistent to provide controls. In the saturated/drained treatment, moisture decreased slowly following drainage due to the high clay content of these soils (27–38% clay; Supplementary Table 1). Soil moisture decreased from 99% WFPS under saturated conditions to 76–80% WFPS at the end of the experiment (Supplementary Fig. 4). During this period, the temporal variation of $CO_2$ production was greater in the saturated/drained treatment than the intermediate treatment and control in all three soils. Overall, there was a slow but consistent increase in $CO_2$ production in the saturated/drained treatment, while $CO_2$ production slowly decreased in the intermediate treatment and was stable in the control (Fig. 1a). By the end of the incubation, cumulative $CO_2$ production was significantly greater ($p < 0.05$) in the saturated/drained treatment than the control in the depression soil, and greater ($p < 0.05$) in the saturated/drained treatment than the intermediate treatment in the ridge soil (Fig. 1b).

Stimulatory effects of increased moisture on C mineralization were even greater after accounting for $CH_4$ fluxes. Substantial $CH_4$ emissions occurred in both the intermediate and saturated treatments after 25 days, exacerbating differences in total C mineralization ($CO_2 + CH_4$) among moisture treatments in all three soils (Fig. 1c). Emissions of $CH_4$ slowly decreased in the intermediate and saturated/drained soils between 82 days and 152 days. Methane emissions accounted for 38% and 30% of total C mineralization in the intermediate and saturated/drained

treatments, respectively, and were negligible (< 0.2% of total C mineralization) in the control due to persistent aerobic conditions. Greater $CH_4$ emissions in the intermediate and saturated/drained treatments resulted in significantly greater cumulative total C mineralization at both 82 and 152 days relative to the control ($p < 0.01$ for all) (Fig. 1d). At 152 days, cumulative total C mineralization in both the saturated/drained ($381 \pm 6$ µmol C g$^{-1}$ soil) and intermediate ($374 \pm 23$ µmol C g$^{-1}$ soil) treatments was significantly higher than in the control ($234 \pm 10$ µmol C g$^{-1}$ soil) ($p < 0.01$ for both) across all three soils (Fig. 1d).

**Relationship between soil moisture and C mineralization**. The traditional model of soil moisture and C mineralization posits that respiration is optimal at intermediate moisture (close to field capacity) and decreases at higher values (Fig. 2), as synthesized in a recent meta-analysis[5]. To compare our observations with this traditional model, we expressed cumulative C mineralization at 25, 82, and 152 days as a function of soil moisture, normalized relative to the field capacity treatment. These dates respectively corresponded with the initiation of $CH_4$ production, initial drainage of the saturated treatment, and the end of the experiment. Trends in cumulative C mineralization with moisture at 25 days closely matched the traditional relationship (Fig. 2), with an optimum at field capacity and decreased values at higher soil moisture. However, at 82 days and 152 days, total C mineralization increased as moisture exceeded field capacity, in contrast to the traditional model. Compared to field capacity, total C mineralization in the intermediate and saturated/drained treatments was respectively increased to 169 and 160% after 82 days and to 160 and 163% after 152 days.

**Sources of mineralized C**. Soils at our study site supported mixed $C_4$–$C_3$ prairie and wetland vegetation over the last 10,000 years[33], and have been cultivated under $C_4$–$C_3$ crop rotations for at least the past 50 years. For this study, soils were collected following

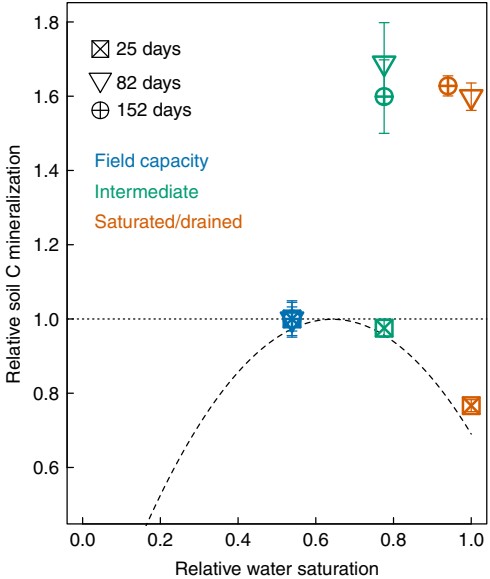

**Fig. 2** Relationships between soil moisture and cumulative C mineralization. Values of soil C mineralization from our study were normalized by the values at field capacity measured at 25, 82, and 152 days, respectively. Soil moisture for the saturated/drained period at 152 days is the mean value over this experiment. The dashed line is the best-fit relationship from Moyano et al.[5]: $SR_H = 3.11\theta_S - 2.42\theta_S^2$, in which $SR_H$ is soil C mineralization and $\theta_S$ is relative water saturation. Each point is the averaged value across the three Mollisols. The error bars indicate s.e.m. ($n = 12$)

corn harvest and amended with corn residues, such that the most recent C inputs had a $C_4$ isotope signature and $C_3$-derived C was older by at least 1 year. The difference in the $\delta^{13}C$ values of total mineralized C ($CO_2 + CH_4$) among the three moisture treatments (Supplementary Figs. 5 and 6 for the $\delta^{13}C$ values of $CO_2$ and $CH_4$, respectively) showed that elevated moisture strongly impacted the source of respired C over time (results normalized by soil C are shown in Supplementary Fig. 7). For the first 25 days, cumulative mineralization of $C_4$-derived C was significantly depressed under the saturated treatment compared with the control and the intermediate treatment ($p < 0.05$ for the difference between the saturated and intermediate treatments in the ridge soil and $p < 0.01$ for the others) (Fig. 3a). Cumulative mineralization from $C_4$-derived C in the intermediate treatment was similar with the control during this period. However, at 82 days, the cumulative $C_4$-derived C mineralization in the depression soil was significantly higher in the saturated and intermediate treatments ($p < 0.01$ for both) than in the control. Mineralization of $C_4$-derived C in the ridge soil also significantly increased in the saturated treatment relative to the control ($p < 0.05$). At the end of the incubation, cumulative C mineralization from $C_4$-derived C was significantly higher in the saturated/drained treatment than in the control in the depression soil, and was significantly greater in the saturated/drained treatment ($p < 0.01$) and the control ($p < 0.05$) than in the intermediate treatment in the footslope soil (Fig. 3b).

Relative to $C_4$-derived C, the mineralization of $C_3$-derived C was even more strongly influenced by soil moisture over the course of the experiment. After the first 25 days, mineralization of $C_3$-derived C was significantly higher in the saturated treatment than the control ($p < 0.05$ for the ridge soil and $p < 0.01$ for the footslope and depression soils), and was also significantly greater in the intermediate treatment vs. the control in the ridge soil ($p < 0.05$) (Fig. 3c). The $C_3$–C contribution to C mineralization in the intermediate and saturated/drained treatments gradually increased over time from 20% at 25 days, to 38% at 82 days, and to 44% at 152 days (Fig. 3d). The contribution of $C_3$–C to C mineralization from control soils varied little over this period: from 14% after 25 days to 21% after 82 days and to 19% after 152 days (Fig. 3d). The total cumulative mineralization of $C_3$-C at the end of this experiment was significantly increased by 356% and 246% in the intermediate and saturated/drained treatments, respectively, relative to the control ($p < 0.01$ for both). Next, we asked whether the changes in DOC following Fe reduction could explain the impacts of moisture on the sources and fluxes of C mineralization.

**Iron reduction and DOC release**. Increased soil moisture in the intermediate and saturated treatments quickly decreased Eh relative to the control across all three soils (Supplementary Fig. 8), indicating decreased $O_2$. Soil Eh dropped from $512 \pm 6$ to $-158 \pm 13$ mV in the intermediate and $-101 \pm 36$ mV in saturated soils during the first 10 days (Supplementary Fig. 8). Iron(II) initially measured $0.2 \pm 0.0$ µmol g$^{-1}$, and after 10 days, decreased Eh corresponded with increased Fe reduction in both the intermediate ($6.0 \pm 3.1$ µmol g$^{-1}$) and saturated ($29.6 \pm 1.9$ µmol g$^{-1}$) treatments relative to the control ($0.5 \pm 0.0$ µmol g$^{-1}$). Iron reduction continued in the intermediate treatment, and Fe(II) measured $72.0 \pm 1.9$ µmol g$^{-1}$ at the end of the experiment (152 days). Fe(II) in the saturated/drained treatment decreased to $11.1 \pm 3.7$ µmol g$^{-1}$ by the end of the experiment as a consequence of increased atmospheric $O_2$ from diffusion and advection following drainage, while Fe(II) measured $0.2 \pm 0.0$ µmol g$^{-1}$ in the control (Supplementary Fig. 9).

An additional companion experiment was conducted with the footslope soil to characterize relationships among Eh, Fe

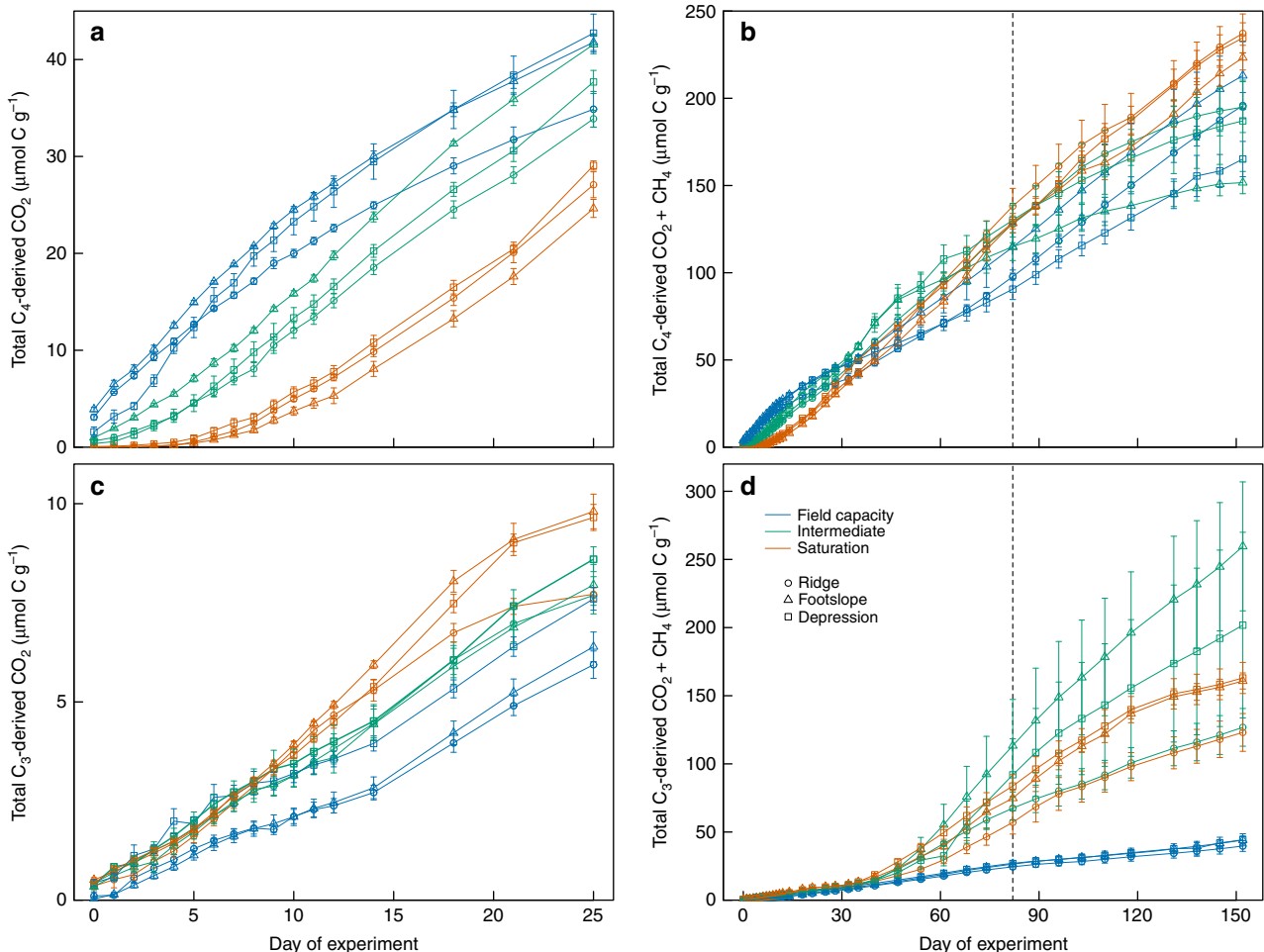

**Fig. 3** Cumulative mineralization of different C sources in three Mollisols incubated under moisture levels at and above field capacity. **a**, **c** Short-term cumulative mineralization of $C_4$-derived C (**a**) and $C_3$-derived C (**c**) respired as $CO_2$; **b**, **d** long-term cumulative mineralization of $C_4$-derived C (**b**) and $C_3$-derived C (**d**) respired as $CO_2$ and $CH_4$. The vertical dashed line indicates when gradual drainage was initiated in the saturated soils. The error bars indicate s.e.m. ($n = 4$)

reduction, pH, and DOC release over 26 days via destructive sampling of 96 additional replicates from the control and saturated treatment (Fig. 4). The footslope soil was chosen because it had hydrological and biogeochemical characteristics (Figs. 1 and 3 and Supplementary Table 1) intermediate between the ridge and depression soils, and similar trends in Eh following saturation (Supplementary Fig. 8). Concentrations of Fe(II) started to increase after 2–4 days in the saturated treatment, and measured $42.4 \pm 0.7 \, \mu\text{mol g}^{-1}$ after 26 days, but Fe(II) did not vary in the control over this period ($0.2 \pm 0.0 \, \mu\text{mol g}^{-1}$ on average) (Fig. 4a). Soil pH varied relatively little over 26 days, increasing from $7.30 \pm 0.06$ to $7.53 \pm 0.03$ in the control and slightly decreasing from $7.08 \pm 0.03$ to $6.87 \pm 0.02$ in the saturated treatment. Concentrations of DOC in the saturated treatment increased in parallel with Fe(II), from $1.28 \pm 0.04$ to $49.04 \pm 1.30$ $\mu\text{mol C g}^{-1}$ after 26 days, and did not vary in the control. The Fe reduction and total DOC release observed after 26-day incubation under saturated conditions were only moderately lower than that obtained by abiotic chemical reduction of replicate subsamples with an inorganic dithionite extraction (Supplementary Fig. 10). The $\delta^{13}\text{C}$ values of DOC (Supplementary Fig. 11) showed that both $C_4$- and $C_3$-derived DOC concentrations increased in the saturated treatment, from $0.76 \pm 0.03$ to $43.82 \pm 0.89 \, \mu\text{mol C g}^{-1}$ after 26 days for $C_4$-derived DOC and from $0.54 \pm 0.04$ to $5.22 \pm 0.62 \, \mu\text{mol C g}^{-1}$ after 26 days for $C_3$-derived DOC (Fig. 4b).

Neither $C_4$- nor $C_3$-derived DOC pools changed with time in the control (Fig. 4b). The fraction of DOC that was readily bioavailable was estimated by incubating DOC samples with a microbial inoculum over a period of five days[34] and measuring $CO_2$ production. The mean percentage of readily bioavailable DOC was similar between the control ($33 \pm 5\%$) and the saturated treatments ($31 \pm 5\%$) (Supplementary Fig. 12).

## Discussion

Our results extend the consensus relationship between soil moisture and C mineralization developed over the last several decades from short-term (days–weeks) incubations, which showed that increased moisture beyond an optimum level (typically near field capacity) suppresses microbial respiration[2, 3, 5]. We show that this framework does not necessarily apply over longer timescales, where prolonged anaerobic conditions and Fe reduction stimulated under elevated moisture can promote the mineralization of protected C (Fig. 2). We found that increasing moisture above field capacity decreased Eh and initially suppressed heterotrophic respiration, consistent with previous work[2, 3, 5]. However, elevated moisture promoted Fe reduction, releasing organic C to solution and altering the sources of mineralized C as indicated by $\delta^{13}\text{C}$ values of $CO_2$ and $CH_4$ (Fig. 3, Supplementary Fig. 1). Mineralization of this previously protected C under

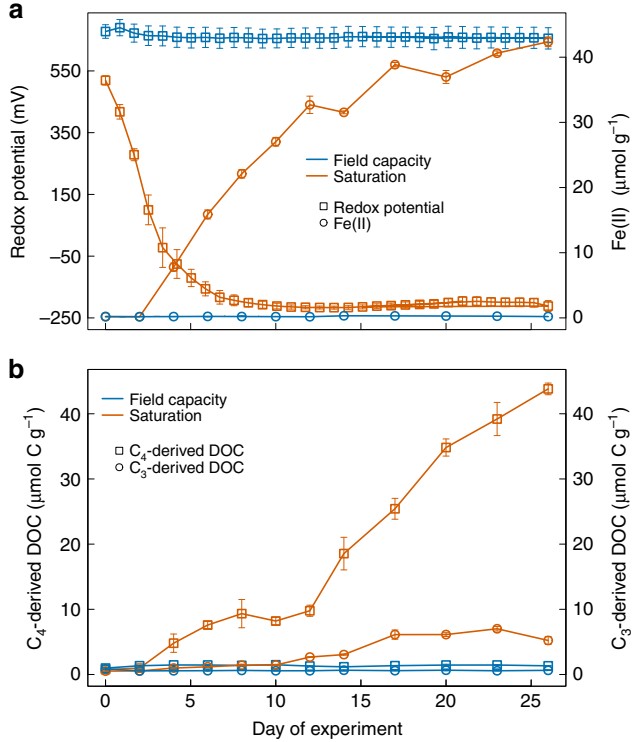

**Fig. 4** Temporal chemical variations in footslope soils incubated under field capacity and saturation. **a** Redox potential (Eh) and ferrous iron (Fe(II)); **b** dissolved organic carbon (DOC) from $C_4$ and $C_3$ sources. The error bars indicate s.e.m. ($n = 4$, except for Eh where $n = 5$)

elevated moisture was sufficient to markedly increase net soil C loss relative to the field-capacity control after several weeks.

Short-range order Fe phases are known to contribute to C stabilization in soils by protecting against microbial mineralization due to C sorption and precipitation of organo-metal complexes[17, 35]. However, portions of this protected C may be released in soluble or colloidal forms following microbial Fe reduction[15, 16, 36]. Previous work suggested that altered pH was largely responsible for DOC release following Fe reduction[19]. However, Pan et al.[21] recently demonstrated that microbial Fe reduction directly released DOC from synthetic C-ferrihydrite associations independently of pH. Similarly, we found that Fe reduction was linked to DOC release from well-buffered mineral soils that experienced little change in pH, precluding a strong influence of soil pH on DOC desorption[15, 19], and thus extending the findings of Pan et al.[21] from synthetic to natural soils.

The $\delta^{13}C$ composition of DOC and mineralized C provide insights into potential sources of Fe-associated C released following reduction. However, DOC is a reactive intermediate pool that reflects the balance between DOC production (via enzymatic and geochemical reactions), and microbial DOC mineralization to $CO_2$ and $CH_4$. Isotope mixing models suggested that the DOC accumulated under saturated conditions in the footslope soil was primarily of $C_4$ origin, but that $C_3$-derived DOC also increased relative to the field capacity control (Fig. 4b). These data help reconcile our finding of increased C mineralization under elevated moisture with traditional theory, which holds that anaerobic conditions suppress C mineralization due to kinetic and thermodynamic constraints[2, 3, 25, 27, 37]. An increase in $C_4$-derived DOC under saturated conditions may partially reflect the accumulation of intermediate decomposition products of $C_4$ litter (added at the beginning of the experiment). This interpretation is consistent with the initial suppression of $C_4$-derived $CO_2$ under

saturated conditions that we observed during the first 25 days. Over the ensuing months, however, much of this $C_4$-derived DOC was ultimately mineralized to $CO_2$ and $CH_4$, along with any additional $C_4$-derived DOC mobilized by Fe reduction. Meanwhile, the increase in $C_3$-derived DOC observed under saturated conditions but not the control (Fig. 4) may have fueled the approximately three-fold increase in $C_3$-derived C mineralization that we ultimately observed in the intermediate and saturated/drained treatments relative to the control (Fig. 3).

In this study system, the $C_3$-derived C was older by at least 1 year relative to the $C_4$ biomass inputs from the most recent growing season and the $C_4$ litter added at the beginning of the experiment. Thus, our observation of increased $C_3$–C mineralization in the intermediate and saturated/drained treatments suggests that reducing conditions can disproportionately lead to the decomposition of older $C_3$–C that was not apparently available to microbial communities under aerobic conditions. A similar phenomenon of increased mineralization of older C under reducing conditions was also recently observed in paddy soils undergoing redox fluctuations[38].

The fact that $C_3$-derived C explained most of the increased C mineralization following Fe reduction under elevated moisture was intriguing, and could potentially be explained by ecohydrological differences among corn and soybeans—the two dominant land covers across the North American Corn Belt. Evapotranspiration is a smaller component of the water budget under soybeans than corn[39], leading to increased soil moisture during the soybean cultivation phase. Increased soil moisture likely increases the potential for Fe reduction and oxidation during soybean cultivation, and the potential sorption or co-precipitation of fresh $C_3$-derived C inputs from soybean residues on short-range-ordered Fe(III) phases that form following sequential Fe reduction and oxidation[40]. This hypothesis does not negate the occurrence of Fe redox cycling and Fe–C complex formation during corn cultivation, however, and the impacts of plant functional types and ecohydrological characteristics on Fe-mediated soil C dynamics merit further investigation.

Together, these results demonstrate that the release of biochemically labile C following Fe reduction and subsequent mineralization as $CO_2$ and $CH_4$ can potentially offset kinetic or thermodynamic constraints that have previously been thought to limit decomposition under anaerobic conditions[25, 27, 37], even as a significant portion of soil C or DOC may remain protected[41] (Supplementary Fig. 12). Analogous to field conditions, soils under intermediate and saturated soil moisture were not isolated from diffusive inputs of atmospheric $O_2$, and contained anaerobic microsites (reflected by decreased Eh, increased Fe(II), and net positive $CH_4$ emissions) within a partially aerobic matrix. Consequently, reactions at the anaerobic/aerobic interface may have been critical in promoting mineralization of organic C released by Fe reduction[25], and in mediating the balance of $CH_4$ production and oxidation following depletion of more thermodynamically favorable electron acceptors within the soil cores[37, 42, 43]. The fractional contribution of $CH_4$ production to total C mineralization in our study (mean of 0.34 in the intermediate and saturated/drained treatments) fell within the reported range of 0.25–0.67 in other methanogenic soils[44]. Omission of $CH_4$ measurements in other studies investigating relationships between moisture and soil C mineralization may have also contributed to differences between our findings and the standard conceptual model (Fig. 2).

In light of our results, one might ask: why is greater SOC content typically observed in anaerobic wetland soils than uplands[45]? The answer may depend on the temporal scales of soil moisture dynamics and the consequent effects on soil C stabilization mechanisms. In consistently flooded soils (i.e., many

months–years), the accumulation of particulate organic C due to suppression of anaerobic lignocellulose decay[27] presumably outweighs any loss of Fe-associated labile C pools solubilized following reductive dissolution. Accordingly, in perennially inundated soils, we often observe a greater contribution of particulate vs. mineral-associated C[46], in contrast to terrestrial soils where mineral-associated C pools often dominate[47]. The importance of elevated moisture in driving Fe-mediated C release will also likely vary as a function of soil characteristics, especially texture. For example, well-drained coarse-textured soils may seldom experience soil saturation[48] and reducing conditions even under substantially elevated precipitation inputs, although this topic merits further study. The clay-rich soils in our study system represent an intermediate between traditional wetlands and well-drained uplands, where Fe-associated organic matter may respond dynamically to redox cycling. We note that similar hydric soils are prevalent across a broad portion of the North American Corn Belt, a region with historically large soil C stocks[49, 50].

To sum up, conventional theoretical linkages between soil C mineralization and soil moisture suggest that high soil moisture limits soil C mineralization because of energetic and enzymatic constraints on microbial activity under short-term (hours–days) anaerobic conditions[3, 27]. This relationship underpins the canonical moisture response functions of heterotrophic respiration in many ecosystem and Earth system models[7, 51–53]. Over timescales of days to several weeks, our data closely matched the traditional unimodal relationship between moisture and C mineralization[7, 52]. However, over longer timescales, these models may greatly underestimate C mineralization when soils experience sustained periods of elevated moisture, given that Fe reduction has been shown to occur in surface soils spanning a broad spectrum of terrestrial ecosystems[22]. The consensus moisture-respiration function underestimated cumulative C mineralization in this study by >50% due to the release and mineralization of protected organic C under elevated moisture (Fig. 2). We suggest that models may significantly underestimate $CO_2$ and $CH_4$ emissions in environments that experience redox oscillations, and this phenomenon may be enhanced under climate change as a consequence of altered frequency and intensity of precipitation[4, 54]. When Fe reduction occurs (e.g., after as little as 2 days of elevated moisture), it can accelerate C loss in mineral soils by facilitating microbial access to previously protected labile C. Ecosystem models could potentially enhance their predictive capacity by explicitly linking soil moisture to Fe-linked biogeochemical mechanisms of C stabilization and release, and including $CH_4$ as well as $CO_2$ production from transient wetland ecosystems.

## Methods

**Site description.** Soils were sampled in November 2015 within a ~400 ha agricultural field in the Walnut Creek watershed (41°75′ N, 93°41′ W) in the Des Moines Lobe geological region of north-central Iowa, USA (Supplementary Fig. 13). Mean monthly temperature ranges from −13.4 °C (January) to 29.4 °C (July). Annual precipitation averages 820 mm[55]. Soils were formed from till following the Wisconsin glaciation and developed under tallgrass prairie and wetland vegetation. Prior to European settlement, ephemeral wetlands covered as much as half of the landscape[56]. Substantial portions of this landscape continue to experience periodic flooding despite drainage infrastructure[29–31]. Our site was cultivated with corn (Zea mays) and soybean (Glycine max) rotated on an annual basis, providing a natural $C_4$–$C_3$ isotope label. Soils were sampled in November 2015 following a corn cultivation phase.

We sampled three common soil series differing in topographic positions and drainage characteristics: Okoboji (depression) are very poorly drained soils at the bottom of topographic depressions, Canisteo (footslope) are moderately poorly drained soils on gentle slopes (0–2°), and Clarion (ridge) are better-drained upland soils. These soils are described as Okoboji mucky silt loam (fine, montmorillonitic, mesic Cumulic Haplaquoll), Canisteo silty clay loam (fine loamy, mixed (calcareous), mesic Typic Haplaquoll) and Clarion loam (fine-loamy, mixed, mesic Cumulic Hapludoll)[55, 57]. We sampled each soil series from each of three separate

$200 \times 200$-m blocks within a ~400 ha field under common management (Supplementary Fig. 13). Six soil cores from each soil series in each block were randomly sampled from 0 to 20 cm (the plow layer) using a 10.2-cm diameter stainless steel auger. Soils from the three blocks were then composited by soil series to generate spatially representative samples at the field scale, which were then split for the incubation experiment. Thus, our incubation samples from each composited soil series are technical replicates rather than spatial replicates, and inference about these soils cannot necessarily be extended beyond the area that was sampled. This approach was chosen given that our focus was to characterize the mean moisture response functions of C mineralization for these soil map units—as opposed to characterizing fine-scale soil spatial variation. The high spatial variation in soil C availability and respiration observed previously in these soils[58] may have otherwise obscured the moisture impacts we sought to test here. These soils are plowed biennially, such that physical disturbance induced by compositing samples and repacking in cores is similar to that which normally occurs at least every 2 years.

**Analysis of $CO_2$ and $CH_4$ production.** We amended soils with finely ground corn leaves (10 mg g$^{-1}$ dry soil) to mimic typical rates of residue incorporation following corn cultivation, which was harvested immediately prior to sampling. The ground corn leaves, along with corn roots produced during the previous growing season, represent the newest inputs of C. Specifically, for each soil core sample, fresh soils (equal to 209.6 g dry soil mass) were mixed well with 2.09 g finely ground corn leaves. The amended soils were then uniformly added to a plastic (butyrate) tube (5 cm diameter, 9.8 cm height) with a polypropylene bottom cap to achieve a bulk density of 1.1 g cm$^{-3}$ and a soil porosity of 58%, representative of field conditions. Soils were incubated at three WFPS levels by adding deionized (18 MΩ) water: field capacity (51% WFPS), intermediate (77% WFPS), and saturation (99% WFPS). We refer to the field capacity treatment as the control. Water was added by injecting a syringe with a stainless steel spinal tap needle to the bottom of the soil core, and slowly moving the needle upward to achieve a uniform distribution of water and displace entrapped air. The initial masses of the soil cores were recorded. Soil moisture was monitored by weighing the soil cores over 3-day intervals throughout the experiment, and water was added as necessary to the surface of the soil cores to reach their initial weight to replace evaporation loss. There were four replicates for each treatment (total $n = 36$). Temperature during the incubations was representative of mean growing season conditions (22–23 °C). The laboratory incubations lasted 152 days. Soils with saturated moisture were allowed to drain slowly after 82 days by perforating the bottom caps of the soil cores with needles, to simulate a typical mid-summer drainage pattern of seasonal wetlands in our area[31]. Soil moisture in the saturated/drained treatment was monitored by recording the mass of the soil cores over 3-day intervals when adding water to the other two treatments to replace evaporation loss.

We measured isotope ratios (δ$^{13}$C) of soil respiration using a tunable diode laser absorption spectrometer (TDLAS, TGA200A, Campbell Scientific, Inc., Logan, Utah) daily for the first two weeks and weekly thereafter. Briefly, $CO_2$-free air was pumped through a manifold with glass jars containing the soil cores, and $CO_2$ production was calculated as the product of steady-state $CO_2$ mole fractions and flow rates. The TGA200A analyzer directly measured $^{12}CO_2$ and $^{13}CO_2$ mole fractions. Isotope ratios were calculated following the convention, in ‰:

$$\delta^{13}C = 1000 \times \left( \frac{[^{13}CO_2]/[^{12}CO_2]}{RPDB} - 1 \right) \quad (1)$$

where RPDB is $^{13}C/^{12}C$ of Vienna PeeDee Belemnite.

After measuring soil $CO_2$ isotope ratios and fluxes by TDLAS, we immediately incubated soil cores in glass jars (485 ml) to measure $CH_4$ production. All jars were re-flushed with $CO_2$-free air and then incubated in the dark for 1 h. Gas samples (20 ml) were collected from each jar via a gas-tight syringe and stored in evacuated 12-ml glass vials. Concentrations of $CH_4$ were analyzed by gas chromatography with a flame ionization detector (GC-2014, Shimadzu, Columbia, MD).

Methane affects the δ$^{13}$C values of $CO_2$ emissions as a consequence of fractionation during both methanogenesis and methane oxidation[59]. To determine δ$^{13}$C of total mineralized C and partition contributions from $C_3$ and $C_4$ biomass, we needed to account for $CH_4$ production from the intermediate and saturated moisture treatments after 25 days. A characteristic steady-state C isotope separation ($\varepsilon_C$) between $CO_2$ and $CH_4$ can be defined for a given soil system that incorporates the combined fractionation from $CH_4$ production and oxidation relative to $CO_2$[59]. To determine $\varepsilon_C$ for our soils, we conducted comprehensive measurements of $^{13}$C of $CH_4$ on four sampling dates after pseudo steady-state production of $CH_4$ had been achieved. Cores were incubated in closed jars for 1 h and two replicate 20 ml gas samples were collected from each jar. After determining $CH_4$ concentrations on one replicate via gas chromatography, δ$^{13}$C of $CH_4$ was measured on the other sample at the UC Davis Stable Isotope Facility using a ThermoScientific Precon concentration unit interfaced to a ThermoScientific Delta V Plus isotope mass spectrometer (ThermoScientifc, Bremen, Germany). We calculated $\varepsilon_C$ as follows:

$$\varepsilon_C = \delta^{13}C_{CH_4} - \delta^{13}C_{CO_2} \quad (2)$$

We assumed a constant value of $\varepsilon_C$ to estimate $\delta^{13}C$ of $CH_4$ for subsequent isotope mass balance calculations. We estimated $\delta^{13}C$ of $CH_4$ for each individual sample at each sampling time as:

$$\delta^{13}C_{CH_4} = \varepsilon_C + \delta^{13}C_{CO_2} \qquad (3)$$

Then, we calculated $\delta^{13}C$ of total mineralized C as:

$$\delta^{13}C_{total} = f \times \delta^{13}C_{CH_4} + (1-f) \times \delta^{13}C_{CO_2} \qquad (4)$$

Here, $f$ is the percentage of $CH_4$ to total mineralized C.

The percent contribution of $C_3$-derived C to total C mineralization ($P_{C_3}$) was then determined using a two-source mixing model:

$$P_{C_3} = 100 \times \frac{\delta^{13}C_{total} - \delta^{13}C_{C_4}}{\delta^{13}C_{C_3} - \delta^{13}C_{C_4}} \qquad (5)$$

By mass balance, $P_{C_4} = 100 - P_{C_3}$. Here, $P_{C_4}$ is the percent contribution of $C_4$-derived C to total C mineralization, and $\delta^{13}C_{C_3}$ and $\delta^{13}C_{C_4}$ represent isotopic end members for $C_3$ and $C_4$ biomass. The bulk $\delta^{13}C$ of added corn residues measured $-12‰$ and was used as the $C_4$ end member. We used $-32‰$ for the $C_3$ end member, which is the lower bound for $C_3$ vegetation identified in a meta-analysis[60]. Finally, we calculated fluxes of mineralized $C_3$ and $C_4$ biomass as the product of $P_{C_3}$ and total mineralized C. The time-integrated $\delta^{13}C$ values of soil-respired $CO_2$ have been shown to be similar to their bulk C sources in pure $C_3$ ecosystems[61, 62], justifying our use of these end-members. Transient C isotope fractionation during decomposition may occur, and is likely be expressed in intermediate pools such as microbial biomass and DOC; however, its overall importance is likely to be minor in comparison with $C_3$–$C_4$ isotopic differences[63].

**Soil biogeochemical analyses**. Subsamples of soils used in incubations for chemical analyses were air dried and passed through a 2-mm sieve. Soil particle size was measured using the hydrometer method. Subsamples for bulk C, N and $\delta^{13}C$ analyses were fumigated with hydrochloric acid to remove any trace carbonates[64] and combusted on an elemental analyzer interfaced with an isotope ratio mass spectrometer (ThermoFinnigan Delta Plus XL, Waltham, MA) at Iowa State University. A ferrozine method optimized for soil extractions[65] was used to analyze citrate/ascorbate-extractable Fe ($Fe_{ca}$), which represents reducible short-range order (oxy)hydroxides and organo-Fe complexes, as well as citrate/dithionite-extractable Fe ($Fe_{cd}$), which represents total free Fe oxides (crystalline and short-range order). The physicochemical properties of the three Mollisols are shown in Supplementary Table 1.

We measured redox potential (Eh) in additional replicate samples across the three soil moisture treatments (3 soil series × 3 soil moisture levels × 2 replicates). The Eh was determined with platinum (Pt) electrodes (Paleo Terra, Amsterdam, Netherlands) that were installed into the middle of the cores at a depth of 5 cm, and connected to a datalogger and multiplexer (Campbell Scientific CR1000 and AM 16/32, Logan, UT). A silver/silver chloride reference electrode was immersed in 2 M KCl and was connected to each soil core via a salt bridge with saturated KCl[66]. Reported Eh values were expressed relative to the standard hydrogen electrode.

To document variations in redox potential (Eh), net Fe reduction, pH, and DOC at higher temporal resolution, we incubated additional replicates of the footslope soil (total $n = 96$) under the field capacity and saturation treatments for 26 days. Ten soil core samples (2 soil moisture treatments × 5 replicates) were used to continuously monitor Eh as described above. Four replicate soil cores per treatment were destructively sampled every 2 days for the first 2 weeks and every 3 days thereafter. Concentrations of Fe(II) were determined colorimetrically using a ferrozine method optimized for soil extractions[65]. Soil pH was measured in 1:1 slurries of soil and deionized water. For DOC measurements, soil subsamples were extracted with nanopure water in a 1:5 soil:solution mass ratio, shaken for 1 h, centrifuged for 10 min at 10,000 rcf, and then filtered through a pre-combusted 0.7-μm glass fiber filter (Whatman GF-F). Concentrations of DOC were analyzed on a Shimadzu TOC-L analyzer (Columbia, MD).

The $\delta^{13}C$ values of DOC were assessed by measuring $\delta^{13}C$ of $CO_2$ produced from oxidation of total DOC by boiling with persulfate[67]. Specifically, 2 ml aliquots of DOC solution were mixed with 1 ml of the oxidizing agent (100 ml $H_2O$ + 4.0 g $K_2S_2O_8$ + 200 μl of 85% $H_3PO_4$) and then acidified to pH < 3 with 8.5% $H_3PO_4$. The solutions were placed in 20-ml glass vials capped with Teflon septa sealed with aluminum crimps, flushed with $CO_2$-free air for 15 min at 500 ml min$^{-1}$, and heated at 100 °C in a water bath for 1 h. The $\delta^{13}C$ values of $CO_2$ oxidized from DOC solutions were measured on the TGA200A analyzer by injection[62] after samples cooled to room temperature. Following oxidation, DOC in the solutions was below detection on the Shimadzu TOC-L, implying quantitative conversion of DOC to $CO_2$. The $CO_2$ produced from blanks containing the oxidizing agent + deionized water was also analyzed to correct the sample $\delta^{13}C$ values. We used isotope mixing models as described above to partition DOC from $C_3$ and $C_4$ sources. This assumes that any $\delta^{13}C$ fractionation associated with sorption and

release of DOC from minerals is small relative to the difference in $\delta^{13}C$ values between $C_3$ and $C_4$ end members[68].

In addition to total DOC, we quantified pools of bioavailable DOC by measuring $CO_2$ produced following 5-day incubations of DOC solutions with a microbial inoculum, according to a standard method[34]. Specifically, 5 ml aliquots of DOC solutions were supplemented with 2 ml nutrient solution (0.1% $(NH_4)_2SO_4$ and 0.1% $KH_2PO_4$) and inoculated with 30 μl unfiltered soil extract that was prepared by shaking fresh footslope soils with 4 mM $CaCl_2$ (1:2) for 10 min[34]. The solutions were placed in 20-ml glass vials capped with butyl rubber septa sealed with aluminum crimps, flushed with $CO_2$-free air for 10 min, and then incubated at 23 °C in the dark. After 5 days, $CO_2$ produced from DOC solutions was measured on the TGA200A analyzer by injection[62]. Negligible $CO_2$ was produced from blank solutions of deionized water amended with the microbial inoculum.

In order to examine the effects of Fe reduction on DOC release, soils were extracted by deionized (18 MΩ) (DI) water and dithionite ($Na_2S_2O_4$) solutions, respectively. There were five replicates for each solution. As $Na_2S_2O_4$ is a strong reductant, it reduces Fe(III) in Fe-containing oxides to Fe(II). We compared the differences in concentrations of Fe(II) and DOC between $Na_2S_2O_4$ extractions and DI water extractions. For $Na_2S_2O_4$ extractions, fresh footslope soils (0.5 g oven dry mass equivalent) were mixed with 0.5 g of $Na_2S_2O_4$ and 30 ml of deionized (18 MΩ) water. For DI water extractions, we similarly mixed the soils and DI water without $Na_2S_2O_4$. The slurries were vortexed for 1 min and then shaken for 16 h. Samples were then centrifuged for 10 min at 10,000 rcf and the supernatant solution was decanted to a clean HDPE bottle for dark storage at 4 °C. Concentrations of DOC were determined on a Shimadzu TOC-L analyzer (Columbia, MD). Concentrations of Fe(II) were measured colorimetrically[65].

**Data analysis**. A mixed-effects model ($n = 4$ per treatment) was used to test for statistical differences among the soils and treatments in the following response variables: instantaneous $CO_2$ production, cumulative $CO_2$ production, instantaneous total C mineralization ($CO_2 + CH_4$), cumulative total C mineralization, and cumulative C mineralization from $C_3$ and $C_4$ sources calculated using the mixing model presented above. A separate mixed-effects model was conducted to analyze the effects of treatments on the above variables for each soil when there were significant interactive effects of soils and treatments. Models included soils and treatments as fixed effects and samples as a random effect (to account for temporal correlation within sampling units) using the lmer function in R[69]. Tukey multiple comparison tests (HSD) were used to test the significant difference among the soils or treatments. All statistical analyses were conducted with the R statistical package[70].

**Data availability**. All the relevant data that support the findings of this study are available from the corresponding author upon request.

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

## Acknowledgements

We thank Frank Hagedorn and two anonymous reviewers for comments that greatly improved the manuscript, and Lucio Reyes and Lindsay Mack for laboratory assistance. This work was funded in part by NSF grant DEB-1457805 and by the Center for Global and Regional Environmental Research at the University of Iowa.

## Author contributions

S.J.H. conceived of this study; W.H. performed research. W.H. and S.J.H. analyzed the data. W.H. and S.J.H. wrote the manuscript.

## Additional information

**Competing interests:** The authors declare no competing financial interests.

