## [Peer Review File · Nature Communications]

Reviewers' comments:

Reviewer #1 (Remarks to the Author):

Hi there

I enjoyed reading your manuscript. It is well written, framed by well-developed hypotheses and backed by a solid experimental design.

Indeed your data does add new knowledge to a growing pool of evidence that soil moisture dynamics may moderate the soil C loss through anaerobic processes co-occurring in otherwise upland or drained minerogenic soils. Especially, addition of CH₄ dynamics as an integral component of C loss under rewetting is in my opinion a major strength of your manuscript and linking it to the biogeochemistry of Fe dynamics points a way forward for field testing and development of models.

I have added my review as an annotated file.

I have two minor issues though I believe you should address in more detail in the introduction and discussion

1. Period of redox change/flooding

You should discuss in more detail what the effect on C mineralization is of shorter periods of redox changes than the 82 days you are using. What is the argument for 82 days? In my opinion the moisture manipulation period is a major factor and an important dynamics to consider when scaling these ideas to a potential field study. In a natural setting, yes 82 days, may be realistic for sustained high moisture content in depression and footslope soils as you simulate here, but is this the case for artificially drained soils or even naturally in the ridge soils? Thus, how representative is 82 days compared to field observations of soil moisture in this landscape - considering also artificially drained soils? Looking at Fig 1 the effect of soil moisture become less and less important a short intervals which is not surprising. So to maximize impact of your study I do believe a more thorough discussion of the flooding period is needed.

2. Soil types and hydrological regime

Convincingly, your results does indicate that added C loss is mainly driven by hydrology and less by soil type. However, this may be because the soils are relatively similar. They are rich in clay, which promotes protection and complexing of organic C with Fe-oxides and allows to sustain high levels of WFPS. How would your results turn out if the same study was carried out in soils with a coarser texture, e.g. more sand and less clay? This would also alter the hydrological regime with shorter

periods of flooding under field conditions (see comment above). Translating your findings in to an operational routine for a model would need to consider the role of soil texture as well as this may constrain rates. In my opinion this is also important to discuss if you want your results to connect to a broader audience.

That being said, your study in isolation makes up for a very good study as is, but you can increase the value by including the abovementioned points.

Reviewer #2 (Remarks to the Author):

The manuscript by Huang & Hall challenges our current understanding that elevated soil moisture and associated anaerobic conditions suppress soil organic C mineralization. Results indicate that under reducing conditions there is even a greater C mineralization than under aerobic ones. As the lost C is old and very likely result from a dissolution of stable mineral associated organic matter, the identified processes may have a great impact on soil's C balance. The relation of soil organic matter and moisture in moister soils is an emerging and so far broadly overlooked topic and the manuscript makes a novel and interesting contribution to it. It may stimulate further research on the duration and susceptibility of SOM stabilization by Fe-oxides.

While soil C models currently apply the concept that C mineralization decreases with water saturation, the relative high C release observed by Huang & Hall could also be deduced from various other studies. For instance, there are a number of studies showing an enhanced DOC release under reducing conditions. Also, Hanke et al (2013, European Journal of Soil Science) found that paddy soils lost more C and substantially older C by soil respiratory processes when they became anaerobic. However, although the findings and the ideas are not entirely novel, Huang & Hall present the first thorough assessment including CO₂ and CH₄ production, stable isotopes, redox potentials, released Fe (II), which will receive a high attention by modeler and experimentalists. Overall, the paper is also nicely written and well set in a theoretical framework.

My main concerns are:

1. The sampling and the set-up of the incubation experiment are somewhat unclear (or sloppy; see more detailed comments below). For instance, it is unclear how large the field was, where the samples had been taken from, if all cores sampled in the field had been mixed to one composite sample and then split for the incubation experiment. This would definitely affect the statistical analysis as there might be no real (field) replicates, but it doesn't affect the story and the mechanisms identified but of course, the quality of the experiment.
2. The $\delta^{13}\text{C}$ value in DOC seems unrealistic and greater than any value reported so far (see specific comments). Also the method how DOC was obtained is unclear.

3. The longer-term perspectives should be discussed more thoroughly. Why do anaerobic soils generally have greater SOC stocks when SOC apparently decompose more rapidly or similarly? See discussion about paddy soils.

Specific comments

Introduction

L. 46 mineral associated C.. 'decompose' over decadal times scale. Her I would rather use the term 'turn over' because at the same time, there will be a C input to mineral associated C and thus a 'replacement'.

Discussion

L. 219 and Figure 4: A $\delta^{13}\text{C}$ value in DOC of -2 ‰ seems very low and seems to be unique (?) to me. Even in the case that all the DOC is released from C4-derived C, then an enrichment by 14 ‰ as compared to the parent material is extraordinary high. Usually, $\delta^{13}\text{C}$ values in DOC are not very different from its parent material (0 to 2 ‰), microbial biomass can be enriched in ^{13}C by 4-6 ‰ (e.g. Kaiser et al., 2001; European Journal of Soil Science). Also clarify that you have measured the $\delta^{13}\text{C}$ in bioavailable DOC and not in total DOC.

L. 229-232 The fact that C3-C appeared to be preferentially associated with Fe oxide phases was intriguing, and could potentially be explained by ecohydrological differences among corn and soybeans. Generally, the average age of mineral-associated C is rather old (>100 y). Therefore, I would not relate it to the different crop types but to the inherent age of the fraction. I suppose that in pre-agricultural time, the area was covered by C3-vegetation (?). Looking at the $\delta^{13}\text{C}$ value in DOC released by Fe-reduction, it is still strongly enriched in ^{13}C with $\delta^{13}\text{C}$ values > -20 ‰.

L. 238 ff Refer to Hanke et al. (2013, European Journal of Soil Science) observing an increased ^{14}C age of soil-respired CO_2 with water-logging in paddy soils.

Methods:

1. The sampling and the processing of the samples in the lab is rather unclear. How large was the field? How many soil cores were taken to have 'spatially representative samples'? Have all cores sampled in the field mixed to one composite sample and then split for the incubation experiment or have the soil from the cores been used directly for the incubation? To me it seems that the set-up seems to be pseudo-replicated as the samples were taken from one field per site (soil type) only and lab replicates have been incubated and analyzed. This does not change the mechanism identified, but at least site (here called soil) effects cannot be analyzed and interpreted in a statistical sense...

2. I think it is necessary to show the $\delta^{13}\text{C}$ measured in soil-respired CO_2 and CH_4 (in a Figure/Table or Text in the supplemental materials). The authors refer to another publication that

$\delta^{13}\text{C}$ values of soil-respired CO_2 is similar to their bulk sources (L. 361). I do not see why this cannot be shown with own data measured (it should!). Also clarify how you calculated the error of your estimates for C3 vs. C4 contribution?

3. L. 374 Concentrations of DOC. How did you extract DOC? Was it filtered? Did you shake the soil?

4. L. 376 Sources of DOC. Clarify that only the sources of biodegradable DOC was measured. Frequently, this is only a fraction of DOC (see Kalbitz et al., 2003, Geoderma). How much was it in this study?

Figures:

Overall, the quality of the graphs could be improved

1. Axis-scale: use rounded numbers (0, 50, 100, ... not 0, 80, 160,...)
2. Report C mineralization in relation to g SOC not g dry soil to allow the reader to get an estimate of how much (which fraction) has been mineralized (without doing the calculation by himself). In this sense, I would also use the similar units in the numerator and denominator (mg $\text{CO}_2\text{-C/gSOC}$ as in most SOC studies or all in mol).
3. The lines look rather messy. Please try to choose thinner lines, smaller symbols or to improve the clarity of the graphs.

Frank Hagedorn

Reviewer #3 (Remarks to the Author):

General comment:

The paper "Elevated moisture stimulates carbon loss from mineral soils by releasing protected organic matter" by Wenjuan Huang & Steven J. Hall presents an experimental study of the carbon (C) loss from mineral soils due to elevated moisture (with soils collected from C-rich former grassland and wetland soils from Iowa, USA). The major claim of the paper is that redox fluctuations can accelerate C loss in mineral soils by facilitating microbial access to occluded C. As a consequence, C mineralization increases at elevated soil moisture.

The paper is concise, clear, well written, and of broad interest for the soil biogeochemistry community. The outcome of this investigation sheds new light into the C-Fe cycles. The fact that redox fluctuations can accelerate C loss in mineral soils by facilitating microbial access to occluded C is novel and can stimulate further research efforts in this direction.

The introduction is clear, however this section can be updated to include works previously done on oxygen limitations. As an example, see the work performed by the group of Porporato. The statistical analyses are appropriate and the experiments are in general well designed, despite some details can be further clarified. For instance, it is not clear which were the redox/oxygen conditions of the saturated samples after they were allowed to drain and how the drainage phase affected the Fe-reduction and C-mineralization processes. Also, the authors did not include information on how the water was added to the soils samples in order to maintain the soil moisture at a constant level, nor the soil moisture values during the drainage phase of the saturated samples. From Figure 2, it seems that the saturated samples drained to a soil moisture value of 0.8, so it did not reach the field capacity during the 70 days of drainage. If this is the case, it would be useful to add this information to the text and provide a desaturation curve of these samples in the supplementary information. Furthermore, more information is needed to clarify some experimental choices. For instance, the authors should explain why the intermediate soils were not allowed to drain, or why some analyses were limited to the footslope soils. Finally, this work is not strictly challenging the relationship between soil moisture and C mineralization found by previous authors, rather it extends this relationship to long-term incubation experiments in which the redox conditions promote the losses of protected C. The authors should add further information on the soil types and biogeochemical conditions for which the proposed C losses are expected to be significant.

Suggestions:

1. Add a scheme of the Fe-C processes described in the supplementary information. Although the authors are not focusing on rates and specific mechanisms, a scheme will help the reader to follow the discussion section.
2. Add the Eh and soil moisture values from day 82 to 152 in the saturated/drained soils.
3. Please, provide further information in the methodological section on how the water content was monitored and how the water content of the samples was adjusted. More details are needed.

Specific comments:

Abstract:

Line 16: 'altered moisture regime' suggests a change in moisture levels, not necessarily an increase. I would rather use 'near saturated conditions' levels or 'under elevated moisture regimes'.

Line 21: 'with recent C4-C inputs little affected' Why were the C4-C inputs little affected? The authors should articulate this point better in the discussion section.

Line 23 'periodic increases in moisture' I would rephrase this sentence given that the loss of C related to iron (Fe) reduction was mainly associated with saturated samples and after a long period of incubation (i.e., 25 days).

Introduction:

Line 27: Please add a reference to support this statement, such as reference [1] of the bibliography provided below (which is already included in the reference list).

Lines 41 and 261-263: This statement is not accurate. Please look at the work performed by Porporato and coauthors [2]. Reference [2] investigates oxygen limitation at elevated soil moisture and quantify the anaerobic respiration rates associated to denitrification. Therefore this citation is relevant.

Lines 49-50: Cite [3] which provides a review of the iron availability in soils.

Line 54: At line 47, the authors state the turnover of mineral-associated C in surface soils is over decadal timescales. However, the authors then say that the release of protected C happens on the weekly to monthly timescales. Please clarify this point.

Line 59 Cite [4] that looked at Iron (III) reduction and phosphorus solubilization in humid tropical forest soils.

Results:

Line 99 and 103: 'Over the first 25 days' Figure 1a displays similar values of CO₂ around day 15, regardless of the treatments. An enlargement of the first 30 days in the SI would help the reader to better follow the result and discussion section.

Line 105 'CO₂ production from the footslope soil was equivalent among the three moisture treatments'. This is not clear from Figure 1a.

Line 110 Explain why the intermediate soil samples were not allowed to slowly drain as the saturated ones. This clarification needs to be included in the paper.

Line 110: It is not clear whether the saturated soils were drained to field capacity? Were the saturated/drained soils aerobic after day 82? No information on changes in oxygen concentration or Eh is provided.

Line 118 Which is the reason of the secondary peak in CO₂ measured at day 30 for both the intermediate treatment and the control? May this increase be associated with the manual addition of water?

Line 125. 'Methane emissions...were negligible (< 0.2 % of total C mineralization) in the control' This result is expected due to the aerobic conditions of the control.

Line 155 'However, after 82 days, the cumulative C₄-derived C mineralization was significantly higher in the saturated treatment ($p < 0.05$) and the intermediate treatment ($p < 0.01$) than in the control' The control of the footslope soil in Figure 3a seems to show values of mineralized C₄-C that are comparable to ones the saturated treatment.

Line 178 The drop observed in Eh for both the intermediate and saturated treatments is in line with the previous comment on Figure 1a.

Line 175: Why did the authors limit the Figure 1 of the Supplementary information to 75 days and did not report the changes in Eh following the drainage of the saturated soils?

Line 185 Why did the authors conduct the companion experiment only on the footslope soils?

Line 176 Why not measure directly oxygen?

Line 176 Which were the oxygen levels in the saturated soils after day 82 (when the soils were allowed to drain)? There are no changes after day 82 in Eh values for the saturated soil, please explain why the redox values did not increase during the drainage phase.

Line 182: Is the lower concentration of Fe(II) for the saturated treatment a consequence of the drainage?

Line 191 Is the DOC in Figure 4b due to C release from C-Fe associations?

Discussion:

Line 201-203 This work is not challenging the relationship between soil moisture and C mineralization found by previous authors (e.g., reference [5]), rather it extends their finding to include long-term incubations for which the redox conditions promote the mineralization of protected C.

Line 250. For the saturated soils, there would be also an advective flux of oxygen due to the drainage of the samples. Please add this information for the sake of physical rigor.

Line 252 Is Figure 3 solely accounting for the CO₂ produced by heterotrophic respiration or also for the oxidation of methane? Did the authors quantify the methane oxidation rates for the saturated/draind samples after day 82?

Line 273. Please quantify the lag at which the C release due to Fe-mediated processes starts. Is the lag associated with a specific range of redox or oxygen conditions?

Methods:

Line 301. Were the ground corn leaves added to analyze the consumption of readily C source and compare it to the mineralization of the occluded C source? Please clarify this point

Figures

Figure 1

Please add an enlargement in Figure 1a (or an extra figure in the Supplementary information) of the first 25 days of the experiment.

Figure 4

In Figure 4 the lag time seems of 12 days. Is this lag consistent with the CO₂ production observed Figure 1? An enlargement of Figure 1 would help the reader.

References

[1] Skopp, J., Jawson, M. D., & Doran, J. W. (1990). Steady-state aerobic microbial activity as a function of soil water content. *Soil Science Society of America Journal*, 54(6), 1619-1625.

[2] Rubol, S., Manzoni, S., Bellin, A., & Porporato, A. (2013). Modeling soil moisture and oxygen effects on soil biogeochemical cycles including dissimilatory nitrate reduction to ammonium (DNRA). *Advances in Water Resources*, 62, 106-124.

[3] Colombo, C., Palumbo, G., He, J. Z., Pinton, R., & Cesco, S. (2014). Review on iron availability in soil: interaction of Fe minerals, plants, and microbes. *Journal of Soils and Sediments*, 14(3), 538-548.

[4] Peretyazhko, T., & Sposito, G. (2005). Iron (III) reduction and phosphorous solubilization in humid tropical forest soils. *Geochimica et Cosmochimica Acta*, 69(14), 3643-3652.

[5] Linn, D. M. & Doran, J. W. Effect of water-filled pore-space on carbon-dioxide and nitrous-oxide production in tilled and nontilled soils. *Soil Sci. Soc. AM. J.* 48, 1267-1272 (1984)

**Reviewers' comments:**

*Reviewer #1 (Remarks to the Author):*

*(1) Period of redox change/flooding*

*You should discuss in more detail what the effect on C mineralization is of shorter*
*periods of redox changes than the 82 days you are using. What is the argument for 82*
*days? In my opinion the moisture manipulation period is a major factor and an*
*important dynamics to consider when scaling these ideas to a potential field study. In*
*a natural setting, yes 82 days, may be realistic for sustained high moisture content in*
*depression and footslope soils as you simulate here, but is this the case for artificially*
*drained soils or even naturally in the ridge soils? Thus, how representative is 82 days*
*compared to field observations of soil moisture in this landscape - considering also*
*artificially drained soils? Looking at Fig 1 the effect of soil moisture become less and*
*less important a short intervals which is not surprising. So to maximize impact of your*
*study I do believe a more thorough discussion of the flooding period is needed.*

**Response:** Agreed. We have added more information on why the soils were saturated
for 82 days in the revised text in several places, and cited data from regional wetland
hydroperiods in van der Valk 2005:

Lines 81-86: “In the most poorly drained soils [in our region], moisture often
increases in spring, remains high for several months, and then decreases in
mid-summer due to increased evapotranspiration³¹. Although soils on footslopes and
depressions are more prone to periodic flooding, soils on ridges also experience
seasonal fluctuations in surface moisture and water table depth³²”

Lines 96-98: “Saturated soils are allowed to drain slowly after 82 days, analogous to
the hydroperiod of seasonal wetlands in our region, while the field capacity and
intermediate treatments remain static”

As stated in the new text, we chose 82 days because it is representative of seasonal
periods of saturation (spring – early summer) that commonly occur in the most poorly
drained soils in our region, and which occurred to an even greater extent prior to
European settlement and drainage infrastructure installation (van der Valk 2005).

The extended period of saturation and drainage provides a useful end member for
challenging the traditional moisture response relationship of soil C mineralization,
which was the focus of this particular study. We agree that shorter-term fluctuations
are also of interest, and these are the focus of ongoing work in our research group. We
have added the following text as justification (lines 98-101):

“Shorter-term moisture fluctuations (days) are also of interest in these ecosystems, but
here we seek to assess biogeochemical impacts of elevated moisture over weeks –
44 months as an end member to challenge conceptual models of heterotrophic activity

developed over shorter timescales.”

Furthermore, we wanted to use a period of flooding that was long enough to achieve
pseudo-steady state Eh and trace gas emissions to use as an end-member for
comparison with the other treatments. We note, however, that our high-frequency gas
flux measurements allowed us to explore the temporal dynamics of microbial
responses to saturation over the entire period of 82 days.

References:

van der Valk, A. G. Water-level fluctuations in North American prairie wetlands.

*Hydrobiologia* **539**, 171-188 (2005)

Khan, F. A. & Fenton, T. E. Saturated zones and soil morphology in a Mollisol catena
of central Iowa. *Soil Sci. Soc. Am. J.* **58**, 1457-1464 (1994)

*(2) Soil types and hydrological regime*

*Convincingly, your results does indicate that added C loss is mainly driven by*
*hydrology and less by soil type. However, this may be because the soils are relatively*
*similar. They are rich in clay, which promotes protection and complexing of organic C*
*with Fe-oxides and allows to sustain high levels of WFPS. How would your results*
*turn out if the same study was carried out in soils with a coarser texture, e.g. more*
*sand and less clay? This would also alter the hydrological regime with shorter*
*periods of flooding under field conditions (see comment above). Translating your*
*findings in to an operational routine for a model would need to consider the role of*
*soil texture as well as this may constrain rates. In my opinion this is also important to*
*discuss if you want your results to connect to a broader audience.*

**Response:** Agreed. As suggested, we have added more discussion on how the patterns
of C mineralization under elevated moisture observed in our study might vary in soils
with a coarser texture:

Lines 340-347: “The importance of elevated moisture in driving Fe-mediated C
release will also likely vary as a function of soil characteristics, especially texture. For
example, well-drained coarse-textured soils may seldom experience soil saturation
and reducing conditions even under substantially elevated precipitation inputs,
although this topic merits further study. The clay-rich soils in our study system
represent an intermediate between traditional wetlands and well-drained uplands,
where Fe-associated organic matter may respond dynamically to redox cycling. We
note that similar hydric soils are prevalent across a broad portion of the North
American Corn Belt, a region with historically large soil C stocks^{49,50}”

*Abstract*

*(3) Line 17 How much above?*

**Response:** Unfortunately, to meet the abstract word limit (150 words) we could not
provide details in the abstract about the WFPS values and their variation over time in

the saturated/drained treatment. We think that the key point for the abstract is that we
observed reducing conditions in both of the treatments with moisture above field
capacity. We have clearly specified the three moisture levels in the last paragraph of
the Introduction (Lines 94-98), as well as in the methods:

“In this study, we assess the effects of soil moisture on soil CO₂ and CH₄ production
and δ¹³C composition in three topographic positions (ridge, footslope, depression) at
three soil moisture levels: field capacity, intermediate, and saturation (51, 77, and 99%
water-filled pore space (WFPS), respectively). Saturated soils are allowed to drain
slowly after 82 days, analogous to the hydroperiod of seasonal wetlands in our region,
while the field capacity and intermediate treatments remain static.”

*(4) Line 20 Please indicate in which measure? Was it on accumulated gaseous C-loss*
*in weight over the 25 day period or comparison of instantaneous CO₂ and CH₄ fluxes*
*between controls and moisture manipulations.*

**Response:** It was the accumulated gaseous C-loss (mass) over the entire period of this
experiment. We have revised “total C mineralization as CO₂ and CH₄” to “cumulative
gaseous C loss as CO₂ and CH₄” to make it clearer. (Line 20)

*(5) Line 20 Just for clarity add that it is carbon isotopes*

**Response:** Agreed. We have revised “stable isotopes” to “stable C isotopes” (Line 20)

*Introduction*

*(6) Line 36 Please make it clear for the reader that this is really diffusion in the soil*
*water. This will make it easier to follow your argumentation below.*

**Response:** Agreed. We have clearly stated that this is C substrate diffusion through
soil water in the sentence. The phrase “oxygen availability and substrate diffusion” has
been revised to “oxygen (O₂) supply from the atmosphere and C substrate diffusion
through soil water” (Lines 35-36).

*(7) Lines 39-41 Still, it would be appropriate with a key ref here addressing this*
*knowledge gap.*

**Response:** Agreed. As this point was also raised by the third Reviewer, we have
revised this sentence and added relevant references (Rubol et al. 2013; McNicol &
Silver 2014). This sentence was revised to “However, the respiratory response at the
other end of the moisture curve, where high moisture limits O₂ availability, has
received less attention^{7,8}” (Lines 39-40)

Rubol, S., Manzoni, S., Bellin, A. & Porporato, A. Modeling soil moisture and oxygen
effects on soil biogeochemical cycles including dissimilatory nitrate reduction to

ammonium (DNRA). *Adv. Water Resour.* **62**, 106-124 (2013).
McNicol, G. & Silver, W. L. Separate effects of flooding and anaerobiosis on soil
greenhouse gas emissions and redox sensitive biogeochemistry. *J. Geophys.*
*Res-Biogeosci.* **119**, 557-566 (2014)

*(8) Line 45 Have a look at this recent paper in Biogeosciences by Zhao et al (2016)*
*Iron-bound organic carbon in forest soils: quantification and characterization*
*doi:10.5194/bg-13-4777-2016. It would serve the purpose of your paper to present for*
*the reader the nature of the Fe-bound organic C in more detail. You mention below*
*the word “labile” which is a key concept underlying your assumptions.*

**Response:** Agreed. We have now cited this paper “Zhao et al. (2016)” and added
more information on the nature of the Fe-bound organic C. We have added the
sentence “For example, hydrophilic and carboxylic C that is readily assimilated by
microbes can be stabilized by Fe oxides via sorption and co-precipitation¹⁰” in this
paragraph. Our data on the bioavailability of DOC that we now report in
Supplemental Fig. 11 and report in the Results on 248-252 also supports this point.

Zhao, Q., et al. Iron-bound organic carbon in forest soils: quantification and
characterization. *Biogeosciences* **13**, 4777-4788 (2016).

*(9) Line 48 Be consistent. Is this OM in general or organic C only?*

**Response:** We have changed “mineral-associated organic matter” to
“mineral-associated organic C” to make it consistent. (Line 50)

*(10) Line 54 Change “protected C” to “organic C bound in Fe-oxide minerals”. You*
*may already here indicate that the proposed mechanism is the breakdown of Fe-org C*
*complexes*

.
**Response:** Agreed. As suggested, we have revised “protected C” to “organic C that is
bound in association with Fe oxide minerals”. (Line 58)

*(11) Lines 58-59 What is significant? Is it the response time, the magnitude or the*
*combination. Would be good for the reader to elucidate more on the refs 10, 18*

**Response:** Good point; here we are referring to the response time. This has been
clarified in the revised text by revising this sentence to “Iron reduction can potentially
occur rapidly (hours – days) following elevated moisture and/or labile C inputs in
many terrestrial soils”. (Lines 62-63)

*(12) Line 61 First time DOC is mentioned. Define and clarify above that it is*
*mineralization of mobilized organic C in the soil water phase, e.g. DOC, that is in*
*focus.*

**Response:** Agreed. Both colloidal and dissolved organic carbon (DOC) may be
released following Fe reduction (Thompson et al. 2006; Buettner et al. 2014;
Hagedorn et al. 2000). So, “Fe-mediated DOC release” was revised to “Fe-mediated
release of colloidal or dissolved organic C (DOC)”. (Line 65)

*(13) L71 This is an important assumption, as the offset of the upstream limitation you*
*mention may be constrained by the structure/properties of released Fe-bound organic*
*C.*

**Response:** Good point; as suggested above, we have specified earlier that labile (i.e.,
hydrophilic and carboxylic moieties) organic C may be bound to Fe and subsequently
released. Please see also our DOC bioavailability assay data in Supplementary Fig. 11
and reported on lines 248-252.

Please also see the revised text on Lines 43-55: “Reactive soil minerals, and iron (Fe)
phases in particular, play a critical role in protecting soil C from microbial
decomposition⁹. For example, hydrophilic and carboxylic C that is readily assimilated
by microbes can be stabilized by Fe oxides via sorption and co-precipitation¹⁰. The
dominant pools of mineral-associated organic C in many surface soils turn over on
decadal timescales and vary with geochemical composition, despite the persistence of
smaller pools of that cycle over centennial to millennial timescales¹¹⁻¹³. However, the
biogeochemical processes that drive the release and subsequent decomposition of
mineral-associated organic C have received less attention (for example, see Keiluweit
*et al.*¹⁴) than mineral protection of C^{9,15}. In particular, protective associations between
Fe mineral phases and soil organic C may be vulnerable to moisture-sensitive redox
dynamics¹⁵⁻¹⁷. For example, organic C which was previously protected over extended
periods by Fe complexation under aerobic conditions could potentially be released
and decomposed following Fe reduction.”

*(14) L86 What does this represent? Can you provide pF values or water filled pore*
*space numbers?*

**Response:** We have provided the values of the water filled pore space for the three
soil moisture levels: three soil moisture levels: field capacity, intermediate, and
saturation (51, 77, and 99% water-filled pore space (WFPS), respectively)”. (Lines
95-97)

*Results*

*(15) Lines 111-112 It would be in place to mention that this is a general observation.*
*However, considerable temporal variation/fluctuation is seen for all three*
*positions/soils which is higher than for the intermediate and control treatments. I*
*think this should be commented as well.*

**Response:** Agreed. We have revised the sentence to “During this period, the temporal
variation of CO₂ production was greater in the saturated/drained treatment than the
intermediate treatment and control in all three soils. Overall, there was a slow but
consistent increase in CO₂ production in the saturated/drained treatment, while CO₂
production slowly decreased in the intermediate treatment and was stable in the
control (Fig. 2a)”. (Lines 143-147)

*(16) Lines 150-153 Was the control and intermediate similar? Please indicate*

**Response:** As suggested, we have indicated the differences in the cumulative
mineralization from C₄-derived C in the revised manuscript. We added the sentence
“Cumulative mineralization from C₄-derived C in the intermediate treatment was
similar with the control during this period.” in the paragraph. (Lines 192-194)

*(17) L153-154 The value of cumulative C4-C mineralization? This is not evident from*
*Fig 3A...*

**Response:** This was unclear, so we have deleted this sentence.

*(18) Lines 193-196 The d13C of DOC started out at +2. What is the reason for this*
*apparent enrichment of DOC?*

**Response:** We note that these values were from rapidly biodegradable DOC, not total
DOC. The enrichment could be due to transient dynamics of C substrate accumulation
and respiratory kinetic fractionation following flooding. We are presently exploring
this phenomenon in greater detail, but as these data may distract from the primary
message of the paper, we have elected not to present them in the revised manuscript.

We therefore used another more conclusive method in this revised manuscript to
measure the C isotope ratio of total DOC (Lines 489-502 of Methods). The δ¹³C
values of total DOC ranged from -20‰ to -13‰ for the saturated treatment and from
252 -23 ‰ to -16‰ for the control, as would be expected from a mixture of C₃- and
253 C₄-derived DOC. The results on the δ¹³C values of total DOC have been presented in
Supplementary Fig. 10. The C₄-derived DOC and C₃-derived DOC have been shown
in Fig. 5 of the main text.

*(19) Line 198 Reverse order of sup Fig 2 & 3*

**Response:** Agreed. We have corrected the order of the supplementary figures.

*Discussion*

*(20) Lines 229-237 You should build this speculation by including if the annual*
*rotation between soy and corn has a role to play in the type of org C bound to Fe*
*oxides. So C3-C sorption/coprecipitation to Fe only occurs under soy bean rotations?*

**Response:** Agreed. We have further discussed how the annual rotation between
soybean and corn play a role in the type of organic C bound to Fe oxides. These
sentences were revised as follows (Lines 305-316):

“The fact that C₃-derived C explained most of the increased C mineralization
following Fe reduction under elevated moisture was intriguing, and could potentially
be explained by ecohydrological differences among corn and soybeans—the two
dominant land covers across the North American Corn Belt. Evapotranspiration is a
smaller component of the water budget under soybeans than corn⁴¹, leading to
increased soil moisture during the soybean cultivation phase. Increased soil moisture
likely increases the potential for Fe reduction and oxidation during soybean
cultivation, and the potential sorption or co-precipitation of fresh C₃-derived C inputs
from soybean residues on short-range-ordered Fe(III) phases that form following
sequential Fe reduction and oxidation⁴². This hypothesis does not negate the
occurrence of Fe redox cycling and Fe-C complex formation during corn cultivation,
however, and the impacts of plant functional types and ecohydrological characteristics
on Fe-mediated soil C dynamics merit further investigation.”

*(21) Lines 246-248 This insight is only possible to get because you include CH₄,*
*which represents the anaerobic pathway of org C decomposition. This is worth*
*mentioning*

**Response:** Agreed. We have revised this sentence to “these results demonstrate that
the release of biochemically labile C following Fe reduction and subsequent
mineralization as CO₂ and CH₄ can potentially offset kinetic or thermodynamic
constraints that have previously been thought to limit decomposition under anaerobic
conditions^{25,27,39}, even as a significant portion of the soil C or DOC pool may remain
protected” (Lines 317-321). Later, we stated that “Omission of CH₄ measurements in
other studies investigating relationships between moisture and soil C mineralization
may have also contributed to differences between our findings and the standard
conceptual model (Fig. 3).” (Lines 329-331)

*(22) Line 268 organic C*

**Response:** Agreed. We have revised “C” to “organic C”. (Line 358)

*(23) Line 269 underestimate gaseous C as CO₂ and CH₄ losses*

**Response:** Agreed. We have revised “underestimate losses” to “underestimate CO₂
and CH₄ emissions”. (Line 359)

*(24) Lines 271-273 ...through the release of Fe-bound organic C*

**Response:** Agreed. We have revised this sentence to “When Fe reduction occurs (e.g.,
after as little as two days of elevated moisture), it can accelerate C loss in mineral
soils by facilitating microbial access to previously protected labile C”. (Lines
361-363)

*(25) Lines 273-275 Yes, if the model would also include CH₄ and CO₂ processes.*

**Response:** Agreed. We have added “and including CH₄ as well as CO₂ production” in
the sentence. (Lines 365-366)

*Methods*

*(26) L303 “residue”*

**Response:** Agreed. We have corrected “reside” to “residue” in the sentence. (Line
404)

*(26) L308 It is not entirely clear to me if this treatment is the field capacity treatment*
*or the field moist conditions as it was under sampling. Please clarify.*

**Response:** This treatment is the field capacity treatment. We have clarified this in the
revised manuscript. (Line 411)

*(27) Fig. 3 As far as I understand the cumulative mineralization includes both CO₂*
*and CH₄. Please clarify this in the caption and figure*

**Response:** Agreed. We have clarified the cumulative mineralization includes both
CO₂ and CH₄ in the caption and figure. Please see Fig. 4.

*(28) Fig. 3 Please clarify in the y-axis title that the mineralized C comprises CH₄ and*
*CO₂.*

**Response:** Agreed. We have clarified that the mineralized C included CH₄ and CO₂ in
the y-axis title. The “Mineralized C₄-C” and “Mineralized C₃-C” have been revised
to “C₄-derived CO₂ + CH₄” and “C₃-derived as CO₂ + CH₄”, respectively. Please see
Fig. 4 in the revised manuscript.

**Reviewer #2 (Remarks to the Author):**

*While soil C models currently apply the concept that C mineralization decreases with*
*water saturation, the relative high C release observed by Huang & Hall could also be*
*deduced from various other studies. For instance, there are a number of studies*
*showing an enhanced DOC release under reducing conditions. Also, Hanke et al*
*(2013, European Journal of Soil Science) found that paddy soils lost more C and*
*substantially older C by soil respiratory processes when they became anaerobic.*

*However, although the findings and the ideas are not entirely novel, Huang & Hall*
*present the first thorough assessment including CO₂ and CH₄ production, stable*
*isotopes, redox potentials, released Fe (II), which will receive a high attention by*
*modeler and experimentalists. Overall, the paper is also nicely written and well set in*
*a theoretical framework.*

**Response:** We really appreciated Dr. Frank Hagedorn's constructive comments and
suggestions. In the first manuscript, we cited many of the earlier studies that showed
release of DOC under reducing conditions (e.g., Thompson et al. 2006, Buettner et al.
2014, Hagedorn et al. 2000, Grybos et al. 2009). The key knowledge gap that we
identify in this paper and address with our experiments is the apparent contradiction
between existing theory, which holds that reducing conditions suppress CO₂
production, and this potentially large release of DOC. To our knowledge, none of the
previous work in this area has addressed this contradiction. Please see revised text in
the Introduction (56-67), which attempts to more clearly drive home this point.

We have added the suggested reference (Hanke et al. 2013) as another important
example to discuss the influence of elevated moisture on the mineralization of
different C sources. (Lines 302-304).

*My main concerns are:*

*(1) The sampling and the set-up of the incubation experiment are somewhat unclear*
*(or sloppy; see more detailed comments below). For instance, it is unclear how large*
*the field was, where the samples had been taken from, if all cores sampled in the field*
*had been mixed to one composite sample and then split for the incubation experiment.*
*This would definitely affect the statistical analysis as there might be no real (field)*
*replicates, but it doesn't affect the story and the mechanisms identified but of course,*
*the quality of the experiment.*

**Response:** We have added more information on the sampling and the set-up of the
incubation experiment. We have provided a map in the Supplementary Information to
show the field sites and the locations of soil sampling (Supplementary Fig. 12). The
sampling scheme was intentionally chosen to allow our experiment to focus on
moisture treatment effects without being confounded by large core-to-core spatial
heterogeneity in C availability, and this is discussed and justified in detail below. Our
sampling details and justification are as follows:

Lines 386-398: "We sampled each soil series from each of three separate 200 x 200-m
blocks within a ~400 ha field under common management (Supplementary Fig. 12).
Six soil cores from each soil series in each block were randomly sampled from 0 – 20
393 cm (the plow layer) using a 10.2-cm diameter stainless steel auger. Soils from the
394 three blocks were then composited by soil series to generate spatially representative
samples at the field scale, which were then split for the incubation experiment. Thus,
our incubation samples from each composited soil series are technical replicates

rather than spatial replicates, and inference about these soils cannot necessarily be
extended beyond the area which was sampled. This approach was chosen given that
our focus was to characterize the mean moisture response functions of C
mineralization for these soil map units—as opposed to characterizing fine-scale soil
spatial variation. The high spatial variation in soil C availability and respiration
observed previously in these soils⁵⁸ may have otherwise obscured the moisture
impacts we sought to test here.”

*(2) The $\delta^{13}\text{C}$ value in DOC seems unrealistic and greater than any value reported so*
*far (see specific comments). Also the method how DOC was obtained is unclear.*

**Response:** We have dealt with this point by conducting additional analyses to
measure $\delta^{13}\text{C}$ values of total DOC (lines 489-502 of Methods), as opposed to the
rapidly biodegradable fraction of DOC that was reported in the first manuscript
version. The $\delta^{13}\text{C}$ values of total DOC are quite consistent with expected values of
DOC derived from a mixture of C₃ and C₄ vegetation. The early enrichment of the
bioavailable DOC we reported in the first manuscript could have been due to transient
kinetic fractionation following flooding. We are presently exploring this phenomenon
in greater detail, but think that this point may distract from the primary messages of
the paper, so we have elected to present the more standard $\delta^{13}\text{C}$ data from total DOC.

The $\delta^{13}\text{C}$ values of total DOC ranged from -20‰ to -13‰ for the saturated treatment
and from -23 ‰ to -16‰ for the control, as would be expected from a mixture of C₃-
and C₄-derived DOC. The results on the $\delta^{13}\text{C}$ values of total DOC have been
presented in the Supplementary Fig. 10. The C₄-derived DOC and C₃-derived DOC
have been shown in Fig. 5 of the main text.

The method for measuring $\delta^{13}\text{C}$ values of total DOC has been added in the revised
manuscript (Lines 489-502): “The $\delta^{13}\text{C}$ values of DOC were assessed by measuring
$\delta^{13}\text{C}$ of CO₂ produced from oxidation of total DOC by boiling with persulfate⁶⁵.
Specifically, 2 mL aliquots of DOC solution were mixed with 1 mL of the oxidizing
agent (100 mL H₂O + 4.9 g K₂S₂O₈ + 200 uL of 85% H₃PO₄) and then acidified to pH
< 3 with 8.5% H₃PO₄. The solutions were placed in 20-ml glass vials capped with
teflon septa sealed with aluminum crimps, flushed with CO₂-free air for 15 min at 500
431 mL min⁻¹, and heated at 100 °C in a water bath for 60 min. The $\delta^{13}\text{C}$ values of CO₂
oxidized from DOC solutions were measured on the TGA200A by injection⁶² after
samples cooled to room temperature. Following oxidation, DOC in the solutions was
below detection on the Shimadzu TOC-L, implying quantitative conversion of DOC
to CO₂. The CO₂ produced from blanks containing the oxidizing agent + deionized
water was also analyzed to correct the sample $\delta^{13}\text{C}$ values.”

We have now specified the method for how DOC was obtained: “For DOC
measurements, soil subsamples were extracted by nanopure water in a 1:5
soil:solution mass ratio, shaken for one hour, centrifuged for 10 min at 10,000 rcf, and

then filtered through a pre-combusted 0.7- μ m glass fiber filter (Whatman GF-F).”
 (Lines 485-487)

*(3) The longer-term perspectives should be discussed more thoroughly. Why do*
 *anaerobic soils generally have greater SOC stocks when SOC apparently decompose*
 *more rapidly or similarly? See discussion about paddy soils.*

**Response:** Agreed, this is an important point. As suggested, the longer-term
 perspectives have now been discussed more thoroughly in the Discussion section:

Lines 332-346: “In light of our results, one might ask: why is greater SOC content
 typically observed in anaerobic wetland soils than uplands⁴⁶? The answer may depend
 on the temporal scales of soil moisture dynamics and the consequent effects on soil C
 stabilization mechanisms. In consistently flooded soils (i.e., many months – years),
 the accumulation of particulate organic C due to suppression of anaerobic
 lignocellulose decay²⁷ presumably outweighs any loss of Fe-associated labile C pools
 solubilized following reductive dissolution. Accordingly, in perennially inundated
 soils, we often observe a greater contribution of particulate vs. mineral-associated C⁴⁷,
 in contrast to terrestrial soils where mineral-associated C pools often dominate⁴⁸. The
 importance of elevated moisture in driving Fe-mediated C release will also likely vary
 as a function of soil characteristics, especially texture. For example, well-drained
 coarse-textured soils may seldom experience soil saturation and reducing conditions
 even under substantially elevated precipitation inputs, although this topic merits
 further study. The clay-rich soils in our study system represent an intermediate
 between traditional wetlands and well-drained uplands, where Fe-associated organic
 matter may respond dynamically to redox cycling.”

*Specific comments*

*Introduction*

*(4) L. 46 mineral associated C.. ‘decompose’ over decadal times scale. Her I would*
 *rather use the term ‘turn over’ because at the same time, there will be a C input to*
 *mineral associated C and thus a ‘replacement’.*

**Response:** Agreed. We have changed “decompose” to “turn over” in the sentence.
 (Line 47)

*Discussion*

*(5) L. 219 and Figure 4: A $\delta^{13}C$ value in DOC of -2 ‰ seems very low and seems to*
 *be unique (?) to me. Even in the case that all the DOC is released from C4-derived C,*
 *then an enrichment by 14 ‰ as compared to the parent material is extraordinary high.*
 *Usually, $\delta^{13}C$ values in DOC are not very different from its parent material (0 to*
 *2 ‰), microbial biomass can be enriched in ^{13}C by 4-6 ‰ (e.g. Kaiser et al., 2001;*
 *European Journal of Soil Science). Also clarify that you have measured the $\delta^{13}C$ in*
 *bioavailable DOC and not in total DOC.*

**Response:** Please see the earlier response on lines 476-493 above; we now report
$\delta^{13}\text{C}$ of total DOC for clarity.

*(6) L. 229-232 The fact that C3-C appeared to be preferentially associated with Fe*
*oxide phases was intriguing, and could potentially be explained by ecohydrological*
*differences among corn and soybeans. Generally, the average age of*
*mineral-associated C is rather old (>100 y). Therefore, I would not relate it to the*
*different crop types but to the inherent age of the fraction. I suppose that in*
*pre-agricultural time, the area was covered by C3-vegetation (?). Looking at the $\delta^{13}\text{C}$*
*value in DOC released by Fe-reduction, it is still strongly enriched in ^{13}C with $\delta^{13}\text{C}$*
*values > -20‰.*

**Response:** There is ongoing controversy as to the apparent ages of mineral associated
C pools. We respectfully disagree with the reviewer on the point regarding the age
distribution of mineral-associated C. While there is indeed a significant amount of
passive mineral-associated C that cycles over centennial to millennial timescales,
rigorous ^{14}C measurements and modeling have shown that large portions of the
mineral-associated C pool cycle on decadal timescales across a wide range of soils
(Trumbore et al. 1995; Baisden et al. 2002; Koarashi et al. 2012; Hall et al. 2015). The
reported centennial mean turnover times of previous studies may often be an artifact
of inappropriate ^{14}C modeling, and the lumping together of mineral-associated C pools
with very different turnover times; see more discussion on this point in Hall et al. 2015,
Biogeosciences 12, 2471–2487.

Trumbore, S. E., Davidson, E. A., Decamargo, P. B., Nepstad, D. C. & Martinelli, L.

511 A. Belowground cycling of carbon in forests and pastures of eastern Amazonia.

*Gloa. Biogeochem. Cy.* **9**, 515-528 (1995).

Baisden, W. T., Amundson, R., Cook, A. C. & Brenner, D. L. Turnover and storage of

C and N in five density fractions from California annual grassland surface soils.

*Glob. Biogeochem. Cy.* **16**, 1117, doi:10.1029/2001GB001822 (2002)

Koarashi, J., Hockaday, W. C., Masiello, C. A. & Trumbore, S. E. Dynamics of

decadal cycling carbon in subsurface soils. *J. Geophys. Res. Biogeosciences*

**117**, G03033, doi:10.1029/2012JG002034 (2012)

Hall, S. J., McNicol, G., Natake, T. & Silver, W. L. Large fluxes and rapid turnover of

mineral-associated carbon across topographic gradients in a humid tropical forest:

insights from paired C-14 analysis. *Biogeosciences* **12**, 2471-2487 (2015)

We have modified the text as follows (Lines 46-48): “The dominant pools of

mineral-associated organic C in many surface soils turn over on decadal timescales

and vary with geochemical composition, despite the persistence of smaller pools of

that cycle over centennial to millennial timescales¹¹⁻¹³.”

Therefore, there is not necessarily any a priori expectation about the C₃ vs C₄

composition of the Fe-associated C, especially given that our site has been under
mixed C₃ – C₄ vegetation for the past several thousand years. We have added
clarifying text about the C₄ and C₃ sources of soil organic C at the site (lines
181-185):

“Soils at our study site supported mixed C₄ – C₃ prairie and wetland vegetation over
the last 10,000 years³³, and have been cultivated under C₄ – C₃ crop rotations for at
least the past 50 years. For this study, soils were collected following corn harvest and
amended with corn residues, such that the most recent C inputs had a C₄ isotope
signature and C₃-derived C was older by at least one year.”

As described above, we have now included the $\delta^{13}\text{C}$ values of total DOC, as well as a
discussion about the factors impacting the $\delta^{13}\text{C}$ composition of DOC (accumulation of
fresh solubilized litter residues in addition to C solubilized following Fe reduction
(lines 281-296):

“Isotope mixing models suggested that the DOC accumulated under saturated
conditions in the footslope soil was primarily of C₄ origin, but that C₃-derived DOC
also increased relative to the field capacity control (Fig. 5). These data help reconcile
our finding of increased C mineralization under elevated moisture with traditional
theory, which holds that anaerobic conditions suppress C mineralization due to kinetic
and thermodynamic constraints^{2,3,25,27,39}. An increase in C₄-derived DOC under
saturated conditions may partially reflect the accumulation of intermediate
decomposition products of C₄ litter (added at the beginning of the experiment). This
interpretation is consistent with the initial suppression of C₄-derived CO₂ under
saturated conditions that we observed during the first 25 days. Over the ensuing
555 months, however, much of this C₄-derived DOC was ultimately mineralized to CO₂
and CH₄, along with any additional C₄-derived DOC mobilized by Fe reduction.
Meanwhile, the increase in C₃-derived DOC observed under saturated conditions but
not the control (Fig. 5) may have fueled the approximately three-fold increase in
C₃-derived C mineralization that we ultimately observed in the intermediate and
saturated/drained treatments relative to the control (Fig. 4).”

*(7) L. 238 ff Refer to Hanke et al. (2013, European Journal of Soil Science) observing*
*an increased 14C age of soil-respired CO2 with water-logging in paddy soils.*

**Response:** Agreed, this is very relevant. We have cited this reference as an example to
discuss the influence of elevated moisture on the mineralization of different C sources.
(Lines 302-304).

*Methods:*

*(8) The sampling and the processing of the samples in the lab is rather unclear. How*
*large was the field? How many soil cores were taken to have ‘spatially representative*
*samples’? Have all cores sampled in the field mixed to one composite sample and*

*then split for the incubation experiment or have the soil from the cores been used*
*directly for the incubation? To me it seems that the set-up seems to be*
*pseudo-replicated as the samples were taken from one field per site (soil type) only*
*and lab replicates have been incubated and analyzed. This does not change the*
*mechanism identified, but at least site (here called soil) effects cannot be analyzed*
*and interpreted in a statistical sense...*

**Response:** According to the reviewer's suggestion, we have specified the field area
and provided more information and justification for the sampling and the sample
processing in the lab (below). The reviewer is correct that the replicates from each
composite soil series are technical replicates suitable to estimate mean values of a
population, not spatial replicates suitable for assessing spatial variability of a
population. This strategy was chosen to minimize the enormous spatial heterogeneity
in C availability that is evident at the scale of individual soil cores at this site
(Cambardella et al. 1994), and thus increase our capacity to detect impacts of
moisture on C mineralization and Fe dynamics, which were the goals of the present
study (as opposed to characterizing spatial variation in the properties of a given soil
series). We respectfully disagree with the statement that "soil" effects cannot be
analyzed in a statistical sense in our study; rather, the scale of inference must be
appropriately qualified to the scale of the composite samples that were taken and
analyzed within our field (see Hurlbert 1984, Ecological Monographs 54: 187-211).
Each composite soil sample represents the mean of a population of 18 cores sampled
from three spatially separate locations of a given soil series throughout the field. To be
sure, we cannot estimate the spatial variability of that population and generalize to the
landscape, but we can certainly test whether the mean of one population differs from
another mean—analogueous to a block effect in ecology. We have emphasized that our
soil series results cannot be necessarily extended beyond the blocks from which they
were taken.

Cambardella, C. A. et al. Field-scale variability of soil properties in central Iowa soils.
Soil Sci. Soc. Am. J. **58**, 1501-1511 (1994).

Hurlbert, S. H. Pseudoreplication and the design of Ecological field experiments. Ecol.
Monogr. **54**, 187-211 (1984)

Please see lines 386-398: "We sampled each soil series from each of three separate
200 x 200-m blocks within a ~400 ha field under common management
(Supplementary Fig. 12). Six soil cores from each soil series in each block were
randomly sampled from 0 – 20 cm (the plow layer) using a 10.2-cm diameter stainless
steel auger. Soils from the three blocks were then composited by soil series to
generate spatially representative samples at the field scale, which were then split for
the incubation experiment. Thus, our incubation samples from each composited soil
series are technical replicates rather than spatial replicates, and inference about these
soils cannot necessarily be extended beyond the area which was sampled. This
approach was chosen given that our focus was to characterize the mean moisture

response functions of C mineralization for these soil map units—as opposed to
characterizing fine-scale soil spatial variation. The high spatial variation in soil C
availability and respiration observed previously in these soils⁵⁸ may have otherwise
obscured the moisture impacts we sought to test here.”

*(9) I think it is necessary to show the $\delta^{13}\text{C}$ measured in soil-respired CO_2 and CH_4*
*(in a Figure/Table or Text in the supplemental materials).*

**Response:** Agreed. As suggested, we have presented the $\delta^{13}\text{C}$ values of soil-respired
CO_2 and CH_4 in the Supplementary Figs 4 and 5.

*The authors refer to another publication that $\delta^{13}\text{C}$ values of soil-respired CO_2 is*
*similar to their bulk sources (L. 361). I do not see why this cannot be shown with own*
*data measured (it should!).*

We respectfully disagree with the reviewer on this minor point. The main point here is
that there is no strong evidence for systematic fractionation between soil organic C
sources and microbial respiration. Although $\delta^{13}\text{C}$ values of soil-respired CO_2 have
been shown to be similar to their sources in pure C_3 ecosystems (see Breecker et al.
2015 and Hall et al., in press), this is not expected in mixed $\text{C}_3 - \text{C}_4$ systems where C_3
and C_4 soil organic C have different ages and turnover times. I.e., we would expect C_4
C losses to be greater in our study system because of the fresh C_4 C that was added at
the beginning of our study.

Please see added text on 469-474: “The time-integrated $\delta^{13}\text{C}$ values of soil-respired
CO_2 have been shown to be similar to their bulk C sources in pure C_3 ecosystems^{61,62},
justifying our use of these end-members. Transient C isotope fractionation during
decomposition may occur, and is likely be expressed in intermediate pools such as
microbial biomass and DOC; however, its overall importance is likely to be minor in
comparison with $\text{C}_3 - \text{C}_4$ isotopic differences⁶³.”

*Also clarify how you calculated the error of your estimates for C_3 vs. C_4*
*contribution?*

We have clarified the original figures by using conventional standard errors for the C_3
vs. C_4 contributions to C mineralization in the revised manuscript.

*(10) L. 374 Concentrations of DOC. How did you extract DOC? Was it filtered? Did*
*you shake the soil?*

Response: As suggested, we have provided more information on the DOC
measurement (Lines 485-487): “For DOC measurements, soil subsamples were
extracted by nanopure water in a 1:5 soil:solution mass ratio, shaken for one hour,
centrifuged for 10 min at 10,000 rcf, and then filtered through a pre-combusted

0.7- μm glass fiber filter (Whatman GF-F).”

*(11) L. 376 Sources of DOC. Clarify that only the sources of biodegradable DOC was*
*measured. Frequently, this is only a fraction of DOC (see Kalbitz et al., 2003,*
*Geoderma). How much was it in this study?*

**Response:** Agreed. Now, we have measured both total DOC and the bioavailable
fraction. The results on the percentage of readily bioavailable DOC have been shown
in the Supplementary Fig. 11. The new text has been added (Lines 251-253): “The
mean percentage of readily bioavailable DOC was similar between the control ($33 \pm$
5%) and the saturated treatments ($31 \pm 5\%$) (Supplementary Fig. 11).

*Figures: Overall, the quality of the graphs could be improved*

*(12) Axis-scale: use rounded numbers (0, 50, 100, ... not 0, 80, 160,...)*

**Response:** Agreed. We have changed to use rounded numbers in the axis-scale in all
the related figures.

*(13) Report C mineralization in relation to g SOC not g dry soil to allow the reader to*
*get an estimate of how much (which fraction) has been mineralized (without doing the*
*calculation by himself). In this sense, I would also use the similar units in the*
*numerator and denominator (mg CO₂-C/gSOC as in most SOC studies or all in mol).*

**Response:** Agreed, but normalizing by total soil organic C pre-supposes that all soil
organic C is equally bioavailable among the three soils, which is not necessarily the
case. Here, we see consistent impacts of moisture treatments on soil C mineralization
across the three soils when expressed on a mass basis but not a soil C basis,
suggesting that the depression soils (with greatest total soil organic C) have a larger
slow or passive pool than the other soil types. For full clarity, as suggested, we have
added figures of C mineralization normalized by SOC in the Supplementary
Information (Supplementary Figs 2 and 6) and used the similar units in the numerator
and denominator (mmol C mol^{-1} SOC) in the related figures.

*(14) The lines look rather messy. Please try to choose thinner lines, smaller symbols*
*or to improve the clarity of the graphs.*

**Response:** Agreed. We have changed to use thinner lines and smaller symbols and
increased the point size of the axis labels to improve the clarity of the graphs.

**Reviewer #3 (Remarks to the Author):**

*General comment:*

*The introduction is clear, however this section can be updated to include works*

*previously done on oxygen limitations. As an example, see the work performed by the*

*group of Porporato.*

**Response:** Good suggestion; we have added the Porporato reference (Rubol et al.
2013) as well as relevant work by McNicol and Silver 2014. (Line 40)

*The statistical analyses are appropriate and the experiments are in general well*
*designed, despite some details can be further clarified. For instance, it is not clear*
*which were the redox/oxygen conditions of the saturated samples after they were*
*allowed to drain and how the drainage phase affected the Fe-reduction and*
*C-mineralization processes.*

**Response:** Good point. On the Figures, we have indicated the period of drainage with
a vertical dashed line, thus the impact of drainage on CO₂ and CH₄ fluxes can be
assessed visually during this period. We have further discussed trends in emissions
following drainage in the text:

“In the saturated/drained treatment, moisture decreased slowly following drainage due
to the high clay content of these soils (27–38% clay; Supplementary Table 1). Soil
moisture decreased from 99% WFPS under saturated conditions to 76-80% WFPS at
the end of the experiment (Supplementary Fig. 3). During this period, the temporal
variation of CO₂ production was greater in the saturated/drained treatment than the
intermediate treatment and control in all three soils. Overall, there was a slow but
consistent increase in CO₂ production in the saturated/drained treatment, while CO₂
production slowly decreased in the intermediate treatment and was stable in the
control (Fig. 2a).”

Unfortunately, we do not have Eh data during the drainage period due to a technical
problem, but we can extrapolate by comparing Fe(II) and Fe(III) data among
treatments at the end of the experiment (Supplemental Fig. 8), where most of the Fe(II)
generated under flooded conditions had oxidized following drainage by the end of the
experiment (Lines):

“Fe(II) in the saturated/drained treatment decreased to $11.1 \pm 3.7 \mu\text{mol g}^{-1}$ by the end
of the experiment as a consequence of increased atmospheric O₂ from diffusion and
advection following drainage, while Fe(II) measured $0.2 \pm 0.0 \mu\text{mol g}^{-1}$ in the control
(Supplementary Fig. 8).”

*Also, the authors did not include information on how the water was added to the soils*
*samples in order to maintain the soil moisture at a constant level, nor the soil*
*moisture values during the drainage phase of the saturated samples. From Figure 2, it*
*seems that the saturated samples drained to a soil moisture value of 0.8, so it did not*
*reach the field capacity during the 70 days of drainage. If this is the case, it would be*
*useful to add this information to the text and provide a desaturation curve of these*
*samples in the supplementary information.*

**Response:** Good points: we have added detailed information about how the water was
added and maintained in the Methods (Lines 411-417):

“Water was added by injecting a syringe with a stainless steel spinal tap needle to the
bottom of the soil core, and slowly moving the needle upward to achieve a uniform
distribution of water and displace entrapped air. The initial masses of the soil cores
were recorded. Soil moisture was monitored by weighing the mass of the soil cores
over three-day intervals throughout the experiment and water was added as necessary
to the surface of the soil cores to reach their initial weight to replace evaporation
loss.”

We have added the requested figure showing the moisture loss over time following
drainage (Supplemental Fig. 3).

Finally, we emphasize that Figure 3 shows the mean moisture content of the
saturated/drained treatment during the 152-day experiment, which we have now stated
in the figure caption: “Soil moisture for the saturated/drained period at 152 days is the
mean value over this experiment.”

*Furthermore, more information is needed to clarify some experimental choices. For*
*instance, the authors should explain why the intermediate soils were not allowed to*
*drain, or why some analyses were limited to the footslope soils.*

**Response:** Good point; please also see our response to Reviewer 1 above (Lines
84-122 in this document) about the justification for the moisture treatments. We
sought to compare a fluctuating hydroperiod representative of seasonal wetlands in
our area with two static hydroperiods (field capacity, and above field capacity but not
saturated) to explore the moisture response of C mineralization under static vs.
saturated/drained moisture. The sustained intermediate and field capacity treatments
provide a useful control for comparison with the saturated/drained treatment.

Lines (97-102): “Saturated soils are allowed to drain slowly after 82 days, analogous
to the hydroperiod of seasonal wetlands in our region, while the field capacity and
intermediate treatments remain static. Shorter-term moisture fluctuations (days) are
also of interest in these ecosystems, but here we seek to assess biogeochemical
impacts of elevated moisture over weeks – months as an end member to challenge
conceptual models of heterotrophic activity developed over shorter timescales.”

We have now clarified that the additional month-long experiment with the footslope
soil was conducted to explore the relationships between Fe reduction, Eh, pH, and
DOC at higher resolution with high replication (total n = 96), which required
extensive destructive analyses. As this soil had characteristics intermediate between
the ridge and depression, we chose it as an illustrative example. We added the

following text (Lines 232-234): “The footslope soil was chosen because it had
hydrological and biogeochemical characteristics (Figs 2 and 4 and Supplementary
Table 1) intermediate between the ridge and depression soils, and similar trends in Eh
following saturation (Supplementary Fig. 7).”

*Finally, this work is not strictly challenging the relationship between soil moisture*
*and C mineralization found by previous authors, rather it extends this relationship to*
*long-term incubation experiments in which the redox conditions promote the losses of*
*protected C. The authors should add further information on the soil types and*
*biogeochemical conditions for which the proposed C losses are expected to be*
*significant.*

**Response:** This is an excellent point. We have rephrased text in the discussion as
follows, and added context as to where this mechanism might be most important:

(Lines 256-261): “Our results extend the consensus relationship between soil moisture
and C mineralization developed over the last several decades from short-term (days –
810 weeks) incubations, which showed that increased moisture beyond an optimum level
(typically near field capacity) suppresses microbial respiration^{2,3,5}. We show that this
framework does not necessarily apply over longer timescales, where anaerobic
conditions and Fe reduction stimulated under elevated moisture can promote the
mineralization of protected C (Fig. 3).”

(Lines 332-347): “In light of our results, one might ask: why is greater SOC content
typically observed in anaerobic wetland soils than uplands⁴⁶? The answer may depend
on the temporal scales of soil moisture dynamics and the consequent effects on soil C
stabilization mechanisms. In consistently flooded soils (i.e., many months – years),
the accumulation of particulate organic C due to suppression of anaerobic
lignocellulose decay²⁷ presumably outweighs any loss of Fe-associated labile C pools
solubilized following reductive dissolution. Accordingly, in perennially inundated
soils, we often observe a greater contribution of particulate vs. mineral-associated C⁴⁷,
in contrast to terrestrial soils where mineral-associated C pools often dominate⁴⁸. The
importance of elevated moisture in driving Fe-mediated C release will also likely vary
as a function of soil characteristics, especially texture. For example, well-drained
coarse-textured soils may seldom experience soil saturation and reducing conditions
even under substantially elevated precipitation inputs, although this topic merits
further study. The clay-rich soils in our study system represent an intermediate
between traditional wetlands and well-drained uplands, where Fe-associated organic
matter may respond dynamically to redox cycling. We note that similar hydric soils
are prevalent across a broad portion of the North American Corn Belt, a region with
historically large soil C stocks^{49,50}.”

*Suggestions:*

(1). *Add a scheme of the Fe-C processes described in the supplementary information.*

*Although the authors are not focusing on rates and specific mechanisms, a scheme*
*will help the reader to follow the discussion section.*

**Response:** Agreed. We have added a schematic of the Fe-C processes described in the
supplementary information. Please see Supplementary Fig. 1.

*(2) Add the Eh and soil moisture values from day 82 to 152 in the saturated/drained*
*soils.*

**Response:** Agreed. We have added the soil moisture values from days 82 – 152
(Supplementary Fig. 3). Unfortunately, due to a technical error, we do not have robust
Eh data after 75 days. However, as discussed above and shown in Supplementary Fig.
8, Fe data from the end of the experiment demonstrate that most of the Fe(II) pool that
accumulated under saturated conditions oxidized following drainage during this
period.

*(3) Please, provide further information in the methodological section on how the*
*water content was monitored and how the water content of the samples was adjusted.*
*More details are needed.*

**Response:** Agreed. Text added as follows:

(Lines 411-417) “Water was added by injecting a syringe with a stainless steel spinal
tap needle to the bottom of the soil core, and slowly moving the needle upward to
achieve a uniform distribution of water and displace entrapped air. The initial masses
of the soil cores were recorded. Soil moisture was monitored by weighing the mass of
the soil cores over three-day intervals throughout the experiment and water was added
as necessary to the surface of the soil cores to reach their initial weight to replace
evaporation loss.”

(Lines 421-424) “Soil moisture in the saturated/drained treatment soil cores was
monitored by recording the mass of the soil cores over three-day intervals when
adding water to the other two treatments to replace evaporation loss.”

*Specific comments: Abstract:*

*(4) Line 16: ‘altered moisture regime’ suggests a change in moisture levels, not*
*necessarily an increase. I would rather use ‘near saturated conditions’ levels or*
*‘under elevated moisture regimes’.*

**Response:** Agreed. We have changed “altered moisture regime” to “under elevated
moisture” (Line 16).

*(5) Line 21: ‘with recent C4-C inputs little affected’ Why were the C4-C inputs little*
*affected? The authors should articulate this point better in the discussion section.*

**Response:** Agreed. For clarity, we have removed this point from the Abstract and
brought it up in greater detail in the Discussion. One hypothesis that we now
articulate in greater detail is that reducing conditions differentially affect the loss of
relatively older C (solubilized by Fe reduction) more than newer C, as recently
observed by others (Hanke et al. 2013). In our ecosystem, Fe-associated C may have a
greater C₃ component because of ecohydrological differences, as described in detail
on lines 305-316.

*(6) Line 23 'periodic increases in moisture' I would rephrase this sentence given that*
*the loss of C related to iron (Fe) reduction was mainly associated with saturated*
*samples and after a long period of incubation (i.e., 25 days).*

**Response:** Agreed; we have rephrased this sentence to “Counter to theory,
elevated moisture may significantly accelerate C losses from mineral soils over weeks
to months—a critical mechanistic deficiency of current Earth system models”.

*Introduction:*

*(7) Line 27: Please add a reference to support this statement, such as reference [1] of*
*the bibliography provided below (which is already included in the reference list).*

**Response:** Agreed. We have added the reference (Skopp et al. 1990) to support the
statement, as well as Linn and Doran 1984.

*(8) Lines 41 and 261-263: This statement is not accurate. Please look at the work*
*performed by Porporato and coauthors [2]. Reference [2] investigates oxygen*
*limitation at elevated soil moisture and quantify the anaerobic respiration rates*
*associated to denitrification. Therefore this citation is relevant.*

**Response:** Agreed. We have revised this sentence and added the suggested reference
(Rubol et al., 2013) in the statement, as well as McNicol and Silver 2014.
Lines 39-40: “However, the respiratory response at the other end of the moisture
curve, where high moisture limits O₂ availability, has received less attention (Rubol et
al. 2013; McNicol and Silver 2014)”. We also have cited the suggested reference in
the discussion section (Line 352).

*(9) Lines 49-50: Cite [3] which provides a review of the iron availability in soils.*

**Response:** Agreed. We have cited the suggested reference (Colombo et al., 2014) at
the end of this sentence (Line 51).

*(10) Line 54: At line 47, the authors state the turnover of mineral-associated C in*
*surface soils is over decadal timescales. However, the authors then say that the*
*release of protected C happens on the weekly to monthly timescales. Please clarify*

*this point.*

**Response:** Agreed. The decadal turnover times are averages that include some
Fe-associated C and some C associated with other mineral phases (see also response
to reviewer 2 on Lines 546-574 above). In our conceptual model (Supplemental
Figure 1), Fe-complexed C may persist for some time but may rapidly be released and
decomposed following Fe reduction. Please see text added on Lines 46-55:

“The dominant pools of mineral-associated organic C in many surface soils turn over
on decadal timescales and vary with geochemical composition, despite the persistence
of smaller pools of that cycle over centennial to millennial timescales¹¹⁻¹³. However,
the biogeochemical processes that drive the release and subsequent decomposition of
mineral-associated organic C have received less attention (for example, see Keiluweit
*et al.*¹⁴) than mineral protection of C^{9,15}. In particular, protective associations between
Fe mineral phases and soil organic C may be vulnerable to moisture-sensitive redox
dynamics¹⁵⁻¹⁷. For example, organic C which was previously protected over extended
periods by Fe complexation under aerobic conditions could potentially be released
and decomposed following Fe reduction.”

*(11) Line 59 Cite [4] that looked at Iron (III) reduction and phosphorus solubilization*
*in humid tropical forest soils.*

**Response:** Agreed. We have cited the suggested reference at the end of this sentence.
(Line 63)

*Results:*

*(12) Line 99 and 103: ‘Over the first 25 days’ Figure 1a displays similar values of*
*CO₂ around day 15, regardless of the treatments. An enlargement of the first 30 days*
*in the SI would help the reader to better follow the result and discussion section.*

**Response:** Agreed. We have added a figure to present the CO₂ production and the
cumulative CO₂ over the first 25 days (Fig. 1) in the main text, which would help the
reader to better follow the results and discussion section. The sentence has been
revised to “During the first 10 days of the experiment, CO₂ production was
consistently depressed in the saturated and the intermediate treatments relative to the
control in all three soils ($p < 0.01$ for all three soils) (Fig. 1a)” (Lines 118-121).

*(13) Line 105 ‘CO₂ production from the footslope soil was equivalent among the*
*three moisture treatments’. This is not clear from Figure 1a.*

**Response:** As also suggested by the second reviewer, we have used thinner lines and
smaller symbols in the figure to make this point clearer. We have clarified that the
CO₂ production “became statistically equivalent” during this period (Line 128).

(14) Line 110 Explain why the intermediate soil samples were not allowed to slowly
drain as the saturated ones. This clarification needs to be included in the paper.

**Response:** Agreed. Please see also the response to reviewer 1. Moisture in the field
capacity and intermediate treatments remained consistent to provide controls relative
to the saturated/drained treatment. We have added the following text:

(Lines 97-99) “Saturated soils are allowed to drain slowly after 82 days, analogous to
the hydroperiod of seasonal wetlands in our region, while the field capacity and
intermediate treatments remain static.”

(Lines 138-139): “Moisture in the field capacity and intermediate treatments remained
consistent to provide controls.”

(15) Line 110: It is not clear whether the saturated soils were drained to field capacity?
Were the saturated/drained soils aerobic after day 82? No information on changes in
oxygen concentration or Eh is provided.

**Response:** We have provided a figure (Supplementary Fig. 3) to show the changes in
soil moisture in saturated/drained treatment after 82 days in the supplementary
information. The following text has been added: (Line 139-142) “In the
saturated/drained treatment, moisture decreased slowly following drainage due to the
high clay content of these soils (27–38% clay; Supplementary Table 1). Soil moisture
decreased from 99% WFPS under saturated conditions to 76-80% WFPS at the end of
the experiment (Supplementary Fig. 3).”

We did not have any data on Eh after 75 days or the O₂ concentration. However, we
have the data on Fe(II) concentrations, which are reflective of O₂ limitation. The Fe(II)
data in our results indicate that the Eh increased after drainage of the saturated soils,
but was still lower than the control.

(16) Line 118 Which is the reason of the secondary peak in CO₂ measured at day 30
for both the intermediate treatment and the control? May this increase be associated
with the manual addition of water?

**Response:** We regularly measured CO₂ at least two hours after we added the water to
the soil cores. All the treatments were treated in the same way for water addition.

Thus, we do not think the peak in CO₂ measured was caused by the manual addition
of water. We speculate that the increase in CO₂ measured at the 32nd and 35th days
may have been associated with microbial dynamics and growth on the added C₄ litter.

Please note that there is increased variability (larger error bars) and greater production
of C₄-derived CO₂ during this period. However, we feel that this is speculative and
tangential to the main points of the paper.

(17) Line 125. 'Methane emissions....were negligible (< 0.2 % of total C
mineralization) in the control' This result is expected due to the aerobic conditions of
the control.

**Response:** Entirely agreed, but still we felt it was necessary to quantify this. The CH₄
emissions were negligible in the control due to the aerobic conditions. We also added
the “due to persistent aerobic conditions” at the end of this sentence to clearly state
the results. (Line 158)

(18) Line 155 'However, after 82 days, the cumulative C₄-derived C mineralization
was significantly higher in the saturated treatment ($p < 0.05$) and the intermediate
treatment ($p < 0.01$) than in the control' The control of the footslope soil in Figure 3a
seems to show values of mineralized C₄-C that are comparable to ones the saturated
treatment.

**Response:** Agreed. The difference in the cumulative C₄-derived C mineralization was
only significant in the depression soils. We missed “in the depression soils” in the
original sentence. Thus, we have corrected this sentence to “at 82 days, the
cumulative C₄-derived C mineralization in the depression soil was significantly higher
in the saturated and intermediate treatments ($p < 0.01$ for both) than in the control.”
(Lines 194-196)

(19) Line 178 The drop observed in Eh for both the intermediate and saturated
treatments is in line with the previous comment on Figure 1a.

Response: This text refers to a drop in Eh under the intermediate and saturated
conditions, which is to be expected in the context of O₂ depletion and Fe reduction
during this period. It also correlates with decreased CO₂, which fits the conventional
model of how soil metabolism responds to anaerobic conditions over the short-term.
Please see text added on lines 260-262.

(20) Line 175: Why did the authors limit the Figure 1 of the Supplementary
information to 75 days and did not report the changes in Eh following the drainage of
the saturated soils?

**Response:** Unfortunately we do not have robust Eh data after 75 days, due to a
technical error. However, we now report trends in moisture for the saturated/drained
treatment during this period (Supplemental Fig. 3), as well as reporting the Fe(II) data
at the end of the experiment. The Fe(II) concentrations in the saturated/drained
treatment at the end of this experiment were much lower than the intermediate
treatment, but slightly higher than the control. The Fe(II) data thus indicate the
maintenance of reducing conditions in the intermediate treatment, and the oxidation of
most Fe(II) produced under saturated conditions following drainage.

(21) Line 185 Why did the authors conduct the companion experiment only on the
footslope soils?

**Response:** As noted in the text, the additional month-long experiment with the
footslope soil was a case study conducted to explore the relationships between Fe
reduction, Eh, pH, and DOC at higher resolution with high replication (total n = 96),
which required extensive destructive analyses. As this soil had characteristics
intermediate between the ridge and depression, we chose it as an illustrative example.
We added the following text: “The footslope soil was chosen because it had
hydrological and biogeochemical characteristics (Figs 2 and 4 and Supplementary
Table 1) intermediate between the ridge and depression soils, and similar trends in Eh
following saturation (Supplementary Fig. 7).” (Lines 232-234)

Due to the expense and labor-intensiveness of this additional experiment, we were
limited to conducting it on one soil type.

(22) Line 176 Why not measure directly oxygen?

**Response:** Eh measurements by Pt electrodes are widely used for characterizing the
redox status of soils and sediments, and both Eh and reduced Fe are correlated with
soil oxygen. Direct measurements of O₂ (with electrodes or via gas wells) would have
been nice to have, but were not critical to the main goals of our study as set up in the
Introduction of the paper.

(23) Line 176 Which were the oxygen levels in the saturated soils after day 82 (when
the soils were allowed to drain)? There are no changes after day 82 in Eh values for
the saturated soil, please explain why the redox values did not increase during the
drainage phase.

**Response:** Eh increased during drainage, as evidenced by the Fe(II) data.

(24) Line 182: Is the lower concentration of Fe(II) for the saturated treatment a
consequence of the drainage?

**Response:** Yes, the lower concentration of Fe(II) for the saturated/drainage treatment
was a consequence of increased atmospheric O₂ during the drainage. We have
rephrased the sentence to make it clear: Iron reduction continued in the intermediate
treatment, and Fe(II) measured $72.0 \pm 1.9 \mu\text{mol g}^{-1}$ at the end of the experiment (152
1095 days). Fe(II) in the saturated/drainage treatment decreased to $11.1 \pm 3.7 \mu\text{mol g}^{-1}$ by the
1096 end of the experiment as a consequence of increased atmospheric O₂ from diffusion and
1097 advection following drainage, while Fe(II) measured $0.2 \pm 0.0 \mu\text{mol g}^{-1}$ in the control
(Supplementary Fig. 8). (Lines 223-228)

(25) Line 191 Is the DOC in Figure 4b due to C release from C-Fe associations?

**Response:** Yes, at least in part, although accumulation of litter decomposition
intermediates also likely contributed. Please see new text on Lines 305-316.

*Discussion:*
(26) Line 201-203 This work is not challenging the relationship between soil moisture
and C mineralization found by previous authors (e.g., reference [5]), rather it extends
their finding to include long-term incubations for which the redox conditions promote
the mineralization of protected C.

**Response:** Agreed. We have revised this sentence to (lines 255-260): “Our results
extend the consensus relationship between soil moisture and C mineralization
developed over the last several decades from short-term (days – weeks) incubations,
which showed that increased moisture beyond an optimum level (typically near field
capacity) suppresses microbial respiration^{2,3,5}. We show that this framework does not
necessarily apply over longer timescales, where anaerobic conditions and Fe
reduction stimulated under elevated moisture can promote the mineralization of
protected C (Fig. 3).”

(27) Line 250. For the saturated soils, there would be also an advective flux of oxygen
due to the drainage of the samples. Please add this information for the sake of
physical rigor.

**Response:** Done; Lines 226-227.

(28) Line 252 Is Figure3 solely accounting for the CO₂ produced by heterotrophic
respiration or also for the oxidation of methane? Did the authors quantify the
methane oxidation rates for the saturated/drained samples after day 82?

**Response:** Please note that we measured net CO₂ and CH₄ fluxes, not gross fluxes,
during the entire experiment (including days 82-152), as is standard for most
incubation studies. We have no explicit quantification of how much of the gross CH₄
that was produced within the core may have been oxidized to CO₂ prior to efflux,
which was beyond the scope of our study. The net fluxes in our study account for both
the CO₂ produced by heterotrophic respiration and the oxidation of methane. We do
quantify the mean contribution of CH₄ to total C mineralization (Lines 326-328):
“The fractional contribution of CH₄ production to total C mineralization in our study
(mean of 0.34 in the intermediate and saturated/drained treatments) fell within the
reported range of 0.25 - 0.67 in other methanogenic soils⁴⁵.”

(29) Line 273. Please quantify the lag at which the C release due to Fe-mediated
processes starts. Is the lag associated with a specific range of redox or oxygen
conditions?

**Response:** Agreed. We have quantified the lag time at which the C release due to
Fe-mediated processes starts. We observe an increase in Fe(II) after as little as two
1147 days, accompanied by increased DOC, as indicated in Fig. 5. Please see new text on
361-363: “When Fe reduction occurs (e.g., after as little as two days of elevated
moisture), it can accelerate C loss in mineral soils by facilitating microbial access to
previously protected labile C.”

We caution to ascribe a specific Eh range to Fe reduction as others have done in the
past, given that Eh is a mixed potential and therefore Eh values for Fe reduction may
vary among systems.

*Methods:*

(30) Line 301. Were the ground corn leaves added to analyze the consumption of
readily available C source and compare it to the mineralization of the occluded C source?

Please clarify this point

**Response:** Yes, the ground corn leaves were added to analyze the mineralization of
readily available C source and compare it to the mineralization of older (by at least
one year) C.

Please see new text in the Results (Lines 183-185): “For this study, soils were
collected following corn harvest and amended with corn residues, such that the most
recent C inputs had a C₄ isotope signature and C₃-derived C was older by at least one
1168 year.”; and new text in the Methods (Lines 403-406): “We amended soils with finely
ground corn leaves (10 mg g⁻¹ dry soil) to mimic typical rates of residue incorporation
following corn cultivation, which was harvested immediately prior to sampling. The
ground corn leaves, along with corn roots produced during the previous growing
season, represent the “newest” inputs of C.”

*Figures*

(31) Figure 1

Please add an enlargement in Figure 1a (or an extra figure in the Supplementary
information) of the first 25 days of the experiment.

**Response:** Agreed. We have added an extra figure to show CO₂ production for the
first 25 days of the experiment in the main text (Fig. 1).

(32) Figure 4

In Figure 4 the lag time seems of 12 days. Is this lag consistent with the CO₂
production observed in Figure 1? An enlargement of Figure 1 would help the reader.

**Response:** We have added an extra figure to show CO₂ production for the first 25
1187 days. The lag time is about 10 days, when the mineralization of the released DOC
following Fe reduction is enough to compensate the depressed decomposition of

recent C4-C inputs. This is also consistent with the CO₂ production observed in
Figure 1a.

*References*

[1] Skopp, J., Jawson, M. D., & Doran, J. W. (1990). Steady-state aerobic microbial
activity as a function of soil water content. *Soil Science Society of America Journal*,
54(6), 1619-1625.

[2] Rubol, S., Manzoni, S., Bellin, A., & Porporato, A. (2013). Modeling soil moisture
and oxygen effects on soil biogeochemical cycles including dissimilatory nitrate
reduction to ammonium (DNRA). *Advances in Water Resources*, 62, 106-124.

[3] Colombo, C., Palumbo, G., He, J. Z., Pinton, R., & Cesco, S. (2014). Review on
iron availability in soil: interaction of Fe minerals, plants, and microbes. *Journal of*
*Soils and Sediments*, 14(3), 538-548.

[4] Peretyazhko, T., & Sposito, G. (2005). Iron (III) reduction and phosphorous
solubilization in humid tropical forest soils. *Geochimica et Cosmochimica Acta*,
69(14), 3643-3652.

[5] Linn, D. M. & Doran, J. W. Effect of water-filled pore-space on carbon-dioxide
and nitrous-oxide production in tilled and nontilled soils. *Soil Sci. Soc. AM. J.* 48,
1267-1272 (1984)

**Response:** We have added all the suggested references in the revised manuscript

REVIEWERS' COMMENTS:

Reviewer #1 (Remarks to the Author):

Hi there

I believe you have done a great job in reviewing your paper and answered the issues raised by all three reviewers.

I have no further comments

Reviewer #2 (Remarks to the Author):

The authors have carefully addressed all the comments by me and the other reviewers, which considerably improved the manuscript. They have added various additional data (e.g. $\delta^{13}\text{C}$ in total DOC) and information strengthening the conclusions and clarifying the setting, sampling as well experimental set-up. I also appreciate the in-depth discussion of the implication of the findings for the longer term soil organic carbon dynamics. The results strongly improve our understanding of soil moisture effects beyond the optimal values for C mineralization and they also indicate that models may underestimate CO_2 and CH_4 emissions in environments that experience redox oscillations. I thus strongly recommend the acceptance of the manuscript

I still have the following minor comments:

L. 70-72: try to condense the sentences about enzymatic activities, reads slightly redundant

L. 95 wording: $\delta^{13}\text{C}$ composition change into ' their $\delta^{13}\text{C}$ values'

L. 102 -107 these are your main results – why do you place them at the end of the introduction - was this a copy-paste mistake?

Fig 1. is part of Fig. 2 – the Figure could be moved to the Supplementals

Frank Hagedorn

Reviewer #3 (Remarks to the Author):

The paper "Elevated moisture stimulates carbon loss from mineral soils by releasing protected organic matter" by Wenjuan Huang & Steven J. Hall presents an experimental study of the carbon (C) loss from mineral soils due to elevated moisture (with soils collected from C-rich former grassland and wetland soils from Iowa, USA). The major claim of the paper is that redox fluctuations can accelerate C loss in mineral soils by facilitating microbial access to occluded C. As a consequence, C mineralization increases at elevated soil moisture. The paper is concise, clear, well written, and of broad interest for the soil biogeochemistry community.

In the revised version of the paper, the authors addressed the Reviewers' comments and reworked significantly the text and the figures. As a result, the manuscript improved significantly in both quality and clarity. I just have some additional comments that the authors should be able to address quickly. Details are listed below.

Detailed comments:

Lines 54-55 of Figure 1 in the Supplementary Information read: 'a significant fraction of soil C may be associated with Fe(III) oxyhydroxides'. Could the authors quantify in the main text how big is (or is expected to be) this fraction in soils and how big this fraction needs to be in order to (potentially) lead to a significant source of anaerobic mineralized C?

Line: 140-141. The composited samples were adjusted to different saturation (51%, 77%, 90%). Given the high clay content, the saturated/drained samples decreased to 76-80% after 82 days of saturation. Waiting for the saturated/drained samples to reach the field capacity would have provided a more meaningful comparison between the saturated/drained treatment and the field capacity control. Please discuss this point.

Line 259: I would change "where anaerobic conditions" to "where prolonged (but not permanent) anaerobic conditions"

Lines 341: Please add a reference (e.g., Hillel 1998, [7])

Lines 386-398: Which was the porosity value of the core samples? Could the authors provide additional details on how the composited samples were prepared and compacted? All the information required to replicate the experiments should be included in the manuscript.

351-355: This sentence needs editing. Models working on a time scale equal/greater than that then one described in the manuscript (as [53]) may underestimate the anaerobic release of C when soils are subjected to intermediate hydro period (and a significant fraction of soil C is associated with Fe(III) oxyhydroxide). However, models developed to describe short time scale incubation experiments as [7] are unlikely going to be affected by the anaerobic release of occluded C. This should be carefully clarified in the text.

Can the authors provide the values of the C mineralization rates under aerobic and anaerobic conditions?

Reference:

Hillel, D. (1998). Environmental soil physics: Fundamentals, applications, and environmental considerations. Academic press.

Response to comments: reviewer comments are shown in italic, and our response in regular font

REVIEWERS' COMMENTS:

Reviewer #1 (Remarks to the Author):

Hi there

I believe you have done a great job in reviewing your paper and answered the issues raised by all three reviewers.

I have no further comments

Response: We thank the reviewer for the previous suggestions.

Reviewer #2 (Remarks to the Author):

1. L. 70-72: try to condense the sentences about enzymatic activities, reads slightly redundant

Response: Agreed. We have condensed the sentence about enzymatic activities. The sentences have been revised to “Activities of soil hydrolytic enzymes that proximately control soil organic matter decomposition are thought to decrease under anaerobic conditions because of decreased enzyme production and inhibition from phenolic substances”. Please see Lines 70-72.

2. L. 95 wording: $\delta^{13}\text{C}$ composition change into ‘ their $\delta^{13}\text{C}$ values ’

Response: Agreed. We have replaced “ $\delta^{13}\text{C}$ composition” to “their $\delta^{13}\text{C}$ values”. Please see Line 93.

3. L. 102 -107 these are your main results – why do you place them at the end of the introduction - was this a copy-paste mistake?

Response: According to the format requirements of Nature Communications (the last paragraph contains a brief summary of both the results and the conclusions (written in present tense)), we placed our main results at the end of the Introduction.

4. Fig 1. is part of Fig. 2 – the Figure could be moved to the Supplementals

Response: Agreed. We have moved the Fig. 1 to the Supplementary Information (now named Supplementary Fig. 2).

Reviewer #3 (Remarks to the Author):

In the revised version of the paper, the authors addressed the Reviewers' comments and reworked significantly the text and the figures. As a result, the manuscript improved significantly in both quality and clarity. I just have some additional comments that the authors should be able to address quickly. Details are listed below.

Detailed comments:

1. Lines 54-55 of Figure 1 in the Supplementary Information read: 'a significant fraction of soil C may be associated with Fe(III) oxyhydroxides'. Could the authors quantify in the main text how big is (or is expected to be) this fraction in soils and how big this fraction needs to be in order to (potentially) lead to a significant source of anaerobic mineralized C?

Response: Agreed. Previous studies (Wagai and Mayer, 2007; Zhao et al., 2016) have showed that Fe-bound organic C could contribute up to 40% of total organic C in soils. We have cited these two references to indicate how large this fraction of soil organic C associated with Fe(III) oxyhydroxides could be in the main text. Please see Lines 52-54.

2. Line: 140-141. The composited samples were adjusted to different saturation (51%, 77%, 90%). Given the high clay content, the saturated/drained samples decreased to 76-80% after 82 days of saturation. Waiting for the saturated/drained samples to reach the field capacity would have provided a more meaningful comparison between the saturated/drained treatment and the field capacity control. Please discuss this point.

Response: This is a relevant point, but our goal here was to assess whether consistent saturation was needed to elicit the observed response in terms of releasing protected C. The saturated/drained treatment C mineralization response shows that even following drainage to intermediate moisture capacity, the net effect on cumulative C mineralization remains the same

3. Line 259: I would change "where anaerobic conditions" to "where prolonged (but not permanent) anaerobic conditions".

Response: Agreed. We have changed "where anaerobic conditions" to "where prolonged anaerobic conditions" in the sentence. Please see Line 257.

4. Lines 341: Please add a reference (e.g., Hillel 1998, [7])

Response: Agreed. We have added the suggested reference in the sentence. Please see Line 341.

5. Lines 386-398: Which was the porosity value of the core samples? Could the authors provide additional details on how the composited samples were prepared and compacted? All the information required to replicate the experiments should be included in the manuscript.

Response: Agreed. We have added information of the soil porosity of the core samples in the text. We also have provided more additional details on how the composited samples were prepared and compacted (Lines 405-409): Specifically, for each soil core sample, fresh soils (equal to 209.6 g dry soil mass) were mixed well with 2.09 g finely ground corn leaves. The amended soils were then uniformly added to a plastic (butyrate) tube (5 cm diameter, 9.8 cm height) with a polypropylene bottom cap to achieve a bulk density of 1.1 g cm^{-3} and a soil porosity of 58%, representative of field conditions.

6. 351-355: This sentence needs editing. Models working on a time scale equal/greater than that then one described in the manuscript (as [53]) may underestimate the anaerobic release of C when soils are subjected to intermediate hydro period (and a significant fraction of soil C is associated with Fe(III) oxyhydroxide). However, models developed to describe short time scale incubation experiments as [7] are unlikely going to be affected by the anaerobic release of occulted C. This should be carefully clarified in the text.

Response: We agree with the overall point. On lines 351-356, we have revised this sentence to “Over timescales of days to several weeks, our data closely matched the traditional unimodal relationship between moisture and C mineralization^{7,52}. However, over longer timescales, these models may greatly underestimate C mineralization when soils experience sustained periods of elevated moisture, given that Fe reduction has been shown to occur in surface soils spanning a broad spectrum of terrestrial ecosystems²².”

7. Can the authors provide the values of the C mineralization rates under aerobic and anaerobic conditions?

Response: These data are readily obtainable from the existing Figures in the manuscript and SI, to the degree that such a distinction between strictly aerobic and anaerobic conditions can be made (see below). In our results (Fig. 1 and Supplementary Figs 2, 3), we showed the instantaneous C mineralization ($\text{CO}_2 + \text{CH}_4$) under three soil moisture levels and the corresponding Eh values for these treatments, which represent the degree of anaerobiosis in each treatment. Please note, however, that the words “aerobic” and “anaerobic” are binary terms that do not adequately describe experiment employed here. We have added the following to clarify (318-321): “Analogous to field conditions, soils under intermediate and saturated soil moisture were not isolated from diffusive inputs of atmospheric O_2 , and contained anaerobic

microsites (reflected by decreased Eh, increased Fe(II), and net positive CH₄ emissions) within a partially aerobic matrix.”

8. *Reference: Hillel, D. (1998). Environmental soil physics: Fundamentals, applications, and environmental considerations. Academic press.*

Response: This reference has been added in the list of References.